# Diffusion-Driven Progressive Target Manipulation for Source-Free Domain Adaptation

**Yuyang Huang[1]***, **Yabo Chen[1]***, **Junyu Zhou[1]**, **Wenrui Dai[1]†**, **Xiaopeng Zhang[2]†**, **Junni Zou[1]†**, **Hongkai Xiong[1]**, **Qi Tian[2]**

[1]Shanghai Jiao Tong University, Shanghai, China    [2]Huawei Inc., Shenzhen, China
{huangyuyang, chenyabo, blabla, daiwenrui, zoujunni, xionghongkai}@sjtu.edu.cn
zxphistory@gmail.com, tian.qi1@huawei.com

*These authors contributed equally to this work. Corresponding authors: Wenrui Dai; Xiaopeng Zhang; Junni Zou.

## Abstract

Source-free domain adaptation (SFDA) is a challenging task that tackles domain shifts using only a pre-trained source model and unlabeled target data. Existing SFDA methods are restricted by the fundamental limitation of source-target domain discrepancy. Non-generation SFDA methods suffer from unreliable pseudo-labels in challenging scenarios with large domain discrepancies, while generation-based SFDA methods are evidently degraded due to enlarged domain discrepancies in creating pseudo-source data. To address this limitation, we propose a novel generation-based framework named Diffusion-Driven Progressive Target Manipulation (DPTM) that leverages unlabeled target data as references to reliably generate and progressively refine a pseudo-target domain for SFDA. Specifically, we divide the target samples into a trust set and a non-trust set based on the reliability of pseudo-labels to sufficiently and reliably exploit their information. For samples from the non-trust set, we develop a manipulation strategy to semantically transform them into the newly assigned categories, while simultaneously maintaining them in the target distribution via a latent diffusion model. Furthermore, we design a progressive refinement mechanism that progressively reduces the domain discrepancy between the pseudo-target domain and the real target domain via iterative refinement. Experimental results demonstrate that DPTM outperforms existing methods by a large margin and achieves state-of-the-art performance on four prevailing SFDA benchmark datasets with different scales. Remarkably, DPTM can significantly enhance the performance by up to 18.6% in scenarios with large source-target gaps.

## 1 Introduction

Deep learning has achieved remarkable success under the independent and identically distributed (i.i.d.) assumption. However, it suffers from significantly degraded performance on out-of-distribution (OOD) data due to domain shifts. Unsupervised domain adaptation (UDA) mitigates this issue by aligning feature distributions between labeled source and unlabeled target domains, but has to access both datasets during adaptation [42, 44]. Source-free domain adaptation (SFDA) considers a more practical but challenging scenario where only the pre-trained source model and unlabeled target data are available [19, 11], and precludes access to source samples during adaptation.

Existing SFDA methods can be primarily classified into non-generation and generation-based methods, with both exhibiting inherent limitations on practical effectiveness. Non-generation methods [23, 57, 61, 35, 38, 37] predominantly rely on pseudo-labels generated by the source model, and categorize them into a small subset of reliable pseudo-labels and a predominant subset of unreliable pseudo-labels

based on certainty metrics [21, 43, 8, 48, 51, 5, 58]. The unreliable pseudo-labels contain substantial label noise, and cannot be easily exploited to extract useful information or infer correct labels through refinement processes. Unfortunately, the amount of label noise is inherently determined by the degree of domain shift between source and target domains [53, 24]. This fundamental limitation severely compromises the effectiveness of non-generation methods in challenging scenarios with large domain gaps, and results in significantly unstable performance across different adaptation tasks. For instance, empirical results [37] show that, when deploying the same source model across different target domains, significant performance discrepancies emerge (*e.g.*, accuracy over 90% for Ar→Cl vs. about 60% for Ar→Pr in the Office-Home dataset). These results underscore the critical sensitivity of non-generation methods to domain shift.

Generation-based SFDA methods primarily operate at the data level [31, 6, 26, 9]. Although these methods could theoretically circumvent the limitation of non-generation methods by avoiding directly using unreliable pseudo-labels, most of them still fail to escape from this domain shift limitation due to the problematic paradigm that generates a pseudo-source domain to convert the SFDA task into a conventional UDA task. The restriction caused by the domain shift between the pseudo-source and target domains is not addressed. Moreover, the generation process often incorporates irrelevant domain features that could further enlarge the discrepancy between the source and target domains for the pseudo-source domain. Consequently, these methods suffer from unsatisfactory performance due to domain shifts.

In this paper, we reveal that existing SFDA methods are fundamentally limited by source-target domain shifts. To break this bottleneck, we resort to a novel generation-based paradigm that directly generates the pseudo-target domain. We propose a Diffusion-Driven Progressive Target Manipulation (DPTM) framework that reliably generates and progressively refines the pseudo target domain to reduce the domain discrepancy from the real target domain and address the fundamental limitation on domain shift for existing methods. The proposed method is shown to achieve remarkable performance gains in challenging DA scenarios with large source-target domain shifts.

To sufficiently and reliably exploit the pseudo-label information, we partition the target data into a trust set and a non-trust set based on the prediction uncertainty of the target model initialized using the source model. For the trust set with low uncertainty, we directly adopt pseudo-labels as supervisory signals for training the target model, following prior works that have established their reliability [21, 43, 8, 48, 51, 5, 58]. Moreover, we also exploit the rich information about the target distribution contained in the potentially unreliable samples from the non-trust set. We uniformly assign a new category label to each of them to prevent potential class imbalance and employ a manipulation strategy to semantically transform each sample toward its newly assigned label while preserving its target-domain features with a latent diffusion model [32]. The manipulated samples simultaneously turn their assigned labels into useful supervisory signals and keep aligned with the target distribution to enhance the adaptation of target models. Furthermore, we propose a Progressive Refinement Mechanism that iteratively refines the pseudo-target domain as well as the target model to progressively reduce the residual label noise due to imperfect pseudo-labeling and the accumulated domain discrepancy caused by the manipulated non-trust set. This significantly diminishes the quantity of non-trust samples and thereby mitigates overall domain shift.

To be concrete, the proposed manipulation strategy of non-trust samples consists of three components. Firstly, as the sampling starting point of diffusion models has been proven to significantly influence the generated image [45, 30, 17, 47, 10], we propose a Target-guided Initialization Mechanism to construct the starting point for sampling by simultaneously considering the target domain features of the non-trust sample and isolating its semantic leakage that might disturb the semantic transformation. Secondly, we propose a Semantic Feature Injection Mechanism that iteratively injects semantics related to the assigned label into the latent throughout the sampling trajectory via DDIM inversion [34, 22] to ensure the semantic transformation without introducing unrelated domain features. Finally, for consistency of manipulated samples with the target distribution, we present a Domain-specific Feature Preservation Mechanism to actively inject target domain features with an adaptively perturbed latent drawn from the original non-trust sample.

Experimental results demonstrate that the proposed method achieves superior performance compared to state-of-the-art (SOTA) methods across four standard SFDA benchmarks of different scales. Remarkably, our method successfully overcomes the limitations of existing methods in challenging domain adaptation scenarios involving large domain shifts. For instance, we achieve a gain of **9.3%**

on D→A and **8.2%** on W→A tasks over the existing SOTA method on the small-scale Office-31 dataset, and a remarkable gain of **18.6%** over SOTA for the Rw→Cl task on the medium-scale Office-Home dataset. On the large-scale DomainNet-126 dataset, we achieve a gain of **24.4%** over the existing generation-based method and **6.3%** over SOTA for the C→P task.

The contributions of this paper are summarized as follows.

- We propose DPTM, a novel framework for Source-free Domain Adaptation (SFDA) that progressively constructs and refines a pseudo-target domain by leveraging unlabeled target data as references with a latent diffusion model.

- We develop a manipulation strategy that leverages Target-guided Initialization, Semantic Feature Injection, and Domain-specific Feature Preservation to semantically transform the unreliable sample toward the newly assigned label while preserving target-domain features.

- We design a Progressive Refinement Mechanism that progressively reduces the domain discrepancy between the pseudo target domain and the real target domain via iterative refinement.

## 2 Related Work

**Source-Free Domain Adaptation.** Source-free domain adaptation (SFDA) methods can be broadly categorized into non-generation and generation-based methods. Non-generation methods [20, 50, 36, 13, 49, 2, 18, 39, 53, 23, 57, 61, 35, 38, 37] mainly employ self-training techniques using pseudo-labels predicted by the source model. However, the inherent unreliability of pseudo-labels caused by source-target domain shifts substantially limits their performance, especially in challenging DA scenarios with significant domain discrepancies. Generation-based methods [23, 57, 61, 35, 38, 37] usually generate pseudo-source domains to convert SFDA into a conventional UDA problem, but their performance remains constrained by the domain shift between the generated pseudo-source and target domains. Different from existing methods, we develop a novel method that directly generates pseudo-target domains and progressively reduces the domain shift between pseudo-target and real-target samples through iterative refinement to overcome the performance limitations.

**Diffusion Models.** Diffusion models [12, 7, 32] have become state-of-the-art in many generative tasks [46, 60, 55, 56, 16, 14, 3]. Their exceptional generation capabilities and pre-trained visual knowledge have been successfully transferred to other vision tasks such as image segmentation [54, 59] and domain generalization [15, 40]. In this work, we leverage diffusion models to facilitate SFDA tasks by semantically transforming unreliable target samples toward their assigned category labels while rigorously preserving their target domain characteristics.

Diffusion models are latent variable generative models defined by a forward and reverse Markov process [12, 7]. The forward process $\{q_t\}_{t\in[0,T]}$ progressively adds Gaussian noise to the data $x_0 \sim q_0(x_0)$ by $q(x_t|x_0) = \mathcal{N}(x_t; \alpha_t x_0, \sigma_t^2 \mathbf{I})$, where the scheduling hyper-parameters $\alpha_t^2 + \sigma_t^2 = 1$. The reverse process $\{p_t\}_{t\in[0,T]}$ gradually removes noise using a learned denoiser $\epsilon_\theta$. Starting from $p(x_T) = \mathcal{N}(\mathbf{0}, \mathbf{I})$, it reconstructs $x_0$ through transitions $p_\theta(x_{t-1}|x_t) = \mathcal{N}(x_{t-1}; x_t - \epsilon_\theta(x_t, t), \sigma_t^2 \mathbf{I})$. Conditional generation is achieved by incorporating condition $y$ into the denoising process as an input to $\epsilon_\theta(x_t, y, t)$. Classifier-free guidance enables conditional generation by combining conditional and unconditional denoising predictions with a guidance scale $\gamma_1$ :

$$\bar{\epsilon}_\theta(x_t, y, t) = (1 + \gamma_1)\epsilon_\theta(x_t, y, t) - \gamma_1 \epsilon_\theta(x_t, \varnothing, t). \tag{1}$$

## 3 Method

### 3.1 Overall Framework

Let $\mathcal{D}_{src}$ a labeled source domain with input space $\mathcal{X}_{src} = \{\mathbf{x}_i^{src}\}_{i=1}^{N_{src}}$ and label space $\mathcal{Y}_{src} = \{y_i^{src}\}_{i=1}^{N_{src}}$, and $\mathcal{D}_{trg}$ an unlabeled target domain with input space $\mathcal{X}_{trg} = \{\mathbf{x}_j^{trg}\}_{j=1}^{N_{trg}}$, where $N_{src}$ and $N_{trg}$ denote the number of samples in the source and target domains, respectively. In SFDA, we first train a source model $\phi_{src} : \mathcal{X}_{src} \to \mathcal{Y}_{src}$ on $\mathcal{D}_{src}$ via supervised learning, and then utilize $\phi_{src}$ and the unlabeled $\mathcal{X}_{trg}$ to learn a target model $\phi_{trg} : \mathcal{X}_{trg} \to \mathcal{Y}_{trg}$ that generalizes well on $\mathcal{D}_{trg}$.

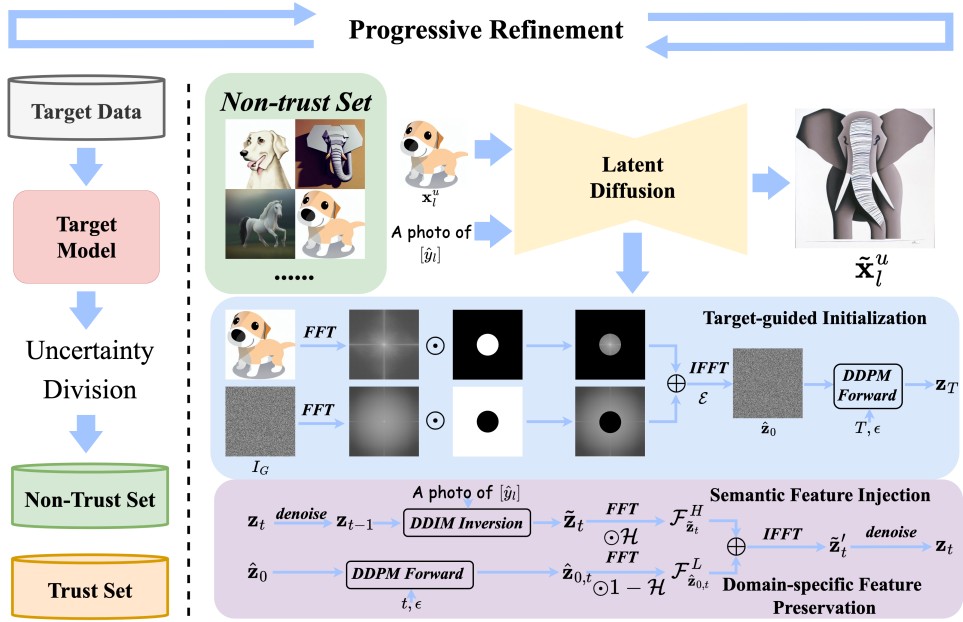

Figure 1: In DPTM, we employ progressive refinement $R$ times: First, we use the target model to make predictions on the target data. Based on each sample's prediction uncertainty, we divide the target data into a trust set and a non-trust set. For the low-uncertainty trust set, we train the target model using pseudo-labels in a supervised manner. For the high-uncertainty non-trust set, we assign a label $\hat{y}_l$ for each sample $\mathbf{x}_l^u$, employ a manipulation strategy that semantically transforms $\mathbf{x}_l^u$ toward class $\hat{y}_l$, while preserving the target-domain features of $\mathbf{x}_l^u$. Our manipulation consists of three components: Target-guided Initialization to obtain an effective sampling starting point, Semantic Feature Injection to convert the semantics of the generated sample to $\hat{y}_l$, and Domain-specific Feature Preservation to maintain the generated sample within the target distribution.

Figure 1 depicts the proposed framework that comprises three key components, including the partition of trust and non-trust sets for unlabeled target data in Section 3.2, manipulation strategy of the non-trust set in Section 3.3, and a progressive refinement mechanism that continuously minimizes the discrepancy between the evolving pseudo-target domain and the real target domain in Section 3.4. We initialize the target model $\phi_{trg}$ with the pre-trained source model $\phi_{src}$. The target domain data is first partitioned into a trust set $\mathcal{V}$ and a non-trust set $\mathcal{U}$ based on prediction uncertainty. For any trust sample $\mathbf{x}_k^v \in \mathcal{V}$, we use the corresponding pseudo-label $y_k^p$ as the supervision signal to train $\phi_{trg}$. The non-trust set undergoes diffusion-based manipulation to produce $\mathcal{U}^m$, which is combined with $\mathcal{V}$ to form the pseudo-target domain $\mathcal{D}_p = \mathcal{V} \cup \mathcal{U}^m$. Finally, we optimize the source model $\phi_{src}$ on this pseudo-target domain in a supervised manner, obtaining a target model $\theta_v$.

## 3.2 Trust and Non-trust Partition for Target Domain

Given any $j$-th unlabeled target data $\mathbf{x}_j^{trg}$ in $\mathcal{X}_{trg}$, we first employ the target model $\phi_{trg}$ to generate the pseudo-label $y_j^p = \arg\max_c p_{\phi_{trg}}(y_c|\mathbf{x}_j^{trg})$, where $p_{\phi_{trg}}(y_c|\mathbf{x}_j^{trg}) = [p(y|\mathbf{x}_j^{trg}; \phi_{trg})]_c$ denotes the probability corresponding to the $c$-th class in the output logits $[p(y|\mathbf{x}_j^{trg}; \phi_{trg})]$ of $\phi_{trg}$. Existing research demonstrates that a small subset of pseudo-labels is trustworthy, while the rest are intrinsically unreliable [21, 43, 8, 48, 51, 5, 58]. The uncertainty can be measured by entropy to distinguish reliable and unreliable pseudo-labels [21, 43, 52, 25]. Therefore, we compute the entropy $H_j^{trg}$ of the target model's prediction $[p(y|\mathbf{x}^{trg}; \phi_{trg})]$ and divide $\mathcal{X}_{trg}$ into trust set $\mathcal{V}$ and non-trust set $\mathcal{U}$ using a threshold $E$. For sample $\mathbf{x}_k^{trg}$ with $H_k^{trg} \leq E$, we consider its pseudo label $y_k^p$ reliable and include $(\mathbf{x}_k^{trg}, y_k^p)$ in $\mathcal{V}$. Otherwise, we solely include the sample in $\mathcal{U}$. Ultimately we obtain the trust set $\mathcal{V} = \{(\mathbf{x}_k^v, y_k^p)\}_{k=1}^{N_v}$ of $N_v$ samples and non-trust set $\mathcal{U} = \{(\mathbf{x}_l^u)\}_{l=1}^{N_u}$ of $N_u$ samples.

## 3.3 Manipulation of Non-trust Set

In this section, we develop a diffusion-based manipulation strategy to further exploit the target-domain information inherently encapsulated in the unreliable pseudo-labels for non-trust samples $\mathbf{x}_l^u \in \mathcal{U}$. For each $\mathbf{x}_l^u \in \mathcal{U}$, we first uniformly assign a category label $\hat{y}_l$ to mitigate potential class imbalance.

$$\hat{y}_l = (l \bmod \lfloor |\mathcal{U}|/C \rfloor), \quad l \in \{1, 2, ..., \lfloor |\mathcal{U}|/C \rfloor \times C\}, \tag{2}$$

where $C$ is the total number of classes. Note that we discard the residual samples with $\lfloor |\mathcal{U}|/C \rfloor \times C < l \leq |\mathcal{U}|$ for class balance. Subsequently, we achieve two objectives via a pre-trained diffusion model at the same time, *i.e.*, i) semantic transformation of $\mathbf{x}_l^u$ toward the specified class $\hat{y}_l$ to convert $\hat{y}_l$ into an effective supervisory signal, and ii) preservation of the target-domain features. The manipulated sample $\tilde{\mathbf{x}}_l^u$ maintains fidelity to the target distribution while exhibiting substantially improved class certainty to allow for converting problematic non-trust samples into useful training instances.

To this end, our diffusion-based manipulation strategy consists of three key components, *i.e.*, i) **Target-guided Initialization** that extracts target domain guidance from $\mathbf{x}_l^u$ to form an effective starting point for the diffusion denoising process, ii) **Semantic Feature Injection** that ensures the designated class $\hat{y}_l$ for generated samples during denoising, and **Domain-specific Feature Preservation** that maintains the generated samples within the target distribution, as elaborated below.

**Target-guided Initialization.** The sampling starting point $x_T$ of diffusion models, particularly its low-frequency components, has been proven to significantly influence the generated image [45, 30, 17, 47, 10]. During inference, the low-frequency components of the generated image and starting point for sampling $x_T$ remain strongly correlated and diffusion models exploit signal leakage from these low-frequency components for image generation [10]. To generate a novel sample from $\mathbf{x}_l^u$ that preserves target domain characteristics while conforming to the newly-assigned category $\hat{y}_l$ of $\mathbf{x}_l^u$, we propose to incorporate the inherent domain-specific features of $\mathbf{x}_l^u$ into the starting point for sampling in diffusion models.

In domain adaptation and generalization, domain-specific features are typically associated with the low-frequency components of $\mathbf{x}_l^u$. Furthermore, to prevent the potential semantic leakage from the high-frequency component of $\mathbf{x}_l^u$, we extract the high-frequency component $\mathcal{F}_{I_G}^H$ from semantically neutral random Gaussian noise $I_G$ and the low-frequency component $\mathcal{F}_{\mathbf{x}_l^u}^L$ from the input image $\mathbf{x}_l^u$ via Fast Fourier Transform ($\mathcal{FFT}$).

$$\mathcal{F}_{\mathbf{x}_l^u}^L = \mathcal{FFT}\left(\mathbf{x}_l^u\right) \odot \mathcal{H}, \quad \mathcal{F}_{I_G}^H = \mathcal{FFT}(I_G) \odot (1 - \mathcal{H}), \tag{3}$$

where $\mathcal{H}$ is a low-pass filter. $\mathcal{F}_{\mathbf{x}_l^u}^L$ and $\mathcal{F}_{I_G}^H$ are combined and inversely transformed via inverse FFT ($\mathcal{IFFT}$) to produce a semantically neutral target-domain pseudo-image $\tilde{\mathbf{x}}_l^u$.

$$\tilde{\mathbf{x}}_l^u = \mathcal{IFFT}\left(\mathcal{F}_{\mathbf{x}_l^u}^L + \mathcal{F}_{I_G}^H\right), \tag{4}$$

$\tilde{\mathbf{x}}_l^u$ is first encoded into the latent space via an encoder $\mathcal{E}$, and then subjected to a $T$-step DDPM forward process to add Gaussian noise. The noisy latent $\mathbf{z}_T$ is used as the starting point for sampling.

$$\hat{\mathbf{z}}_0 = \mathcal{E}(\tilde{\mathbf{x}}_l^u), \quad \mathbf{z}_T = \sqrt{\alpha_T}\hat{\mathbf{z}}_0 + \sqrt{1 - \alpha_T}\boldsymbol{\epsilon}, \quad \boldsymbol{\epsilon} \sim \mathcal{N}(\mathbf{0}, \mathbf{I}). \tag{5}$$

**Semantic Feature Injection.** For our task, the sampling starting point $\mathbf{z}_T$ derived from (5) may inherently lack sufficient semantic relevance to $\hat{y}_l$ due to the following reasons. First, we construct the high-frequency components of $\mathbf{z}_T$ using a semantically neutral Gaussian noise image, which carries no $\hat{y}_l$-related information. Secondly, although we isolate the high-frequency components of $\mathbf{x}_l^u$, weak semantic leakage from $\mathbf{x}_l^u$ may persist, potentially conflicting with $\hat{y}_l$. Consequently, we may fail to semantically transform $\mathbf{x}_l^u$ to $\hat{y}_l$ even with a large guidance scale $\gamma_1$ according to [22]. To address this, we present semantic feature injection as below.

During denoising, at each timestep $t$, we adopt a zigzag self-reflection operation following [22]. We first denoise the latent $\mathbf{z}_t$ via the latent diffusion model to obtain $\mathbf{z}_{t-1}$, and then yield the refined latent $\tilde{\mathbf{z}}_t$ by injecting $\hat{y}_l$-related semantic information into $\mathbf{z}_{t-1}$ with DDIM inversion [34].

$$\tilde{\mathbf{z}}_t = \sqrt{\frac{\alpha_t}{\alpha_{t-1}}}\mathbf{z}_{t-1} + \sqrt{\alpha_t}\left(\sqrt{\frac{1}{\alpha_t} - 1} - \sqrt{\frac{1}{\alpha_{t-1}} - 1}\right)\tilde{\epsilon}_\theta\left(\mathbf{z}_{t-1}, \hat{y}_l, t-1\right) \tag{6}$$

$$\tilde{\epsilon}_\theta\left(\mathbf{z}_{t-1}, \hat{y}_l, t-1\right) = (1 + \gamma_2)\epsilon_\theta\left(\mathbf{z}_{t-1}, \hat{y}_l, t-1\right) - \gamma_2\epsilon_\theta\left(\mathbf{z}_{t-1}, \varnothing, t-1\right),$$

where $\gamma_2$ is the inversion guidance scale. According to (6), semantic alignment with $\hat{y}_l$ is considered for the latents throughout the sampling trajectory. However, since DDIM inversion could introduce unrelated domain features, the latents could deviate from the target distribution when accumulating $\hat{y}_l$-aligned semantic information. To address this, we selectively extract the high-frequency components carrying the accumulated semantic information and discard the low-frequency components harboring domain artifacts from $\tilde{\mathbf{z}}_t$ rather than directly leveraging $\tilde{\mathbf{z}}_t$.

$$\mathcal{F}_{\tilde{\mathbf{z}}_t}^H = \mathcal{FFT}(\tilde{\mathbf{z}}_t) \odot (1 - \mathcal{H}). \tag{7}$$

$\mathcal{F}_{\tilde{\mathbf{z}}_t}^H$ is used as high-frequency semantics to aggregate with target domain specific features.

**Domain-specific Feature Preservation.** To better align with the target distribution, we combine the high-frequency semantic features $\mathcal{F}_{\tilde{\mathbf{z}}_t}^H$ in the latents by DDIM inversion in (7) with the target domain specific features at each denoising timestep $t$. The domain-specific features are primarily encoded in the low-frequency components of samples from the target domain. To adapt to the time-varying noise level in $\mathbf{z}_t$, we perturb the clean latent $\hat{\mathbf{z}}_0$ in (5) via the DDPM forward process with $t$-step Gaussian noise to generate $\hat{\mathbf{z}}_{0,t} = \sqrt{\alpha_t}\hat{\mathbf{z}}_0 + \sqrt{1-\alpha_t}\boldsymbol{\epsilon}$ for timestep $t$. The low-frequency domain-specific features $\mathcal{F}_{\hat{\mathbf{z}}_{0,t}}^L$ are extracted from $\hat{\mathbf{z}}_{0,t}$ and combined with $\mathcal{F}_{\tilde{\mathbf{z}}_t}^H$ to obtain enhanced latent $\tilde{\mathbf{z}}_t'$ that simultaneously preserves $\hat{y}_l$-aligned high-frequency semantics and embeds target-domain low-frequency features to produce $\mathbf{z}_{t-1}$.

$$\tilde{\mathbf{z}}_t' = \mathcal{IFFT}\left(\mathcal{F}_{\hat{\mathbf{z}}_{0,t}}^L + \mathcal{F}_{\tilde{\mathbf{z}}_t}^H\right), \quad \mathcal{F}_{\hat{\mathbf{z}}_{0,t}}^L = \mathcal{FFT}(\hat{\mathbf{z}}_{0,t}) \odot \mathcal{H}. \tag{8}$$

### 3.4 Progressive Refinement Mechanism

We design a progressive refinement mechanism to iteratively refine the pseudo-target domain for $R$ iterations to further optimize the target model. When optimized solely on a fixed pseudo-target domain, the target model could be affected by the trust set $\mathcal{V}$ inevitably contains residual label noise due to imperfect pseudo-labeling, and the manipulated non-trust set $\mathcal{U}^m$ gradually accumulates domain discrepancy in sample generation. Therefore, for any $r$-th ($r = 1, \cdots, R$) refinement iteration, we update the trust set $\mathcal{V}^r$ to correct inaccurate pseudo-labels and reduce the size of the non-trust set $\mathcal{U}^r$ for decreasing domain discrepancy. The target model $\phi_{trg}^r$ re-partitions the target data into updated trust set $\mathcal{V}^{(r+1)}$ and non-trust set $\mathcal{U}^{(r+1)}$. $\mathcal{U}^{(r+1)}$ is then further manipulated according to Section 3.3 to generate $\mathcal{U}^{m,(r+1)}$ for constructing the refined pseudo-target domain $\mathcal{D}_p^{(r+1)} = \mathcal{V}^{(r+1)} \cup \mathcal{U}^{m,(r+1)}$. The updated target model $\phi_{trg}^{(r+1)}$ is obtained by fine-tuning $\theta_v^{(r)}$ on $\mathcal{D}_p^{(r+1)}$. We empirically find in Figure 3 that, during progressive refinement, $\mathcal{V}^{(r+1)}$ provides more accurate pseudo-labels than $\mathcal{V}^{(r)}$ with $|\mathcal{V}^{(r+1)}| > |\mathcal{V}^{(r)}|$ such that $|\mathcal{U}^{m,(r+1)}| < |\mathcal{U}^{m,(r)}|$ to reduce the size of manipulated non-trust set and decrease the domain discrepancy in the pseudo-target domain. Compared with $\phi_{trg}^{(r)}$, $\phi_{trg}^{(r+1)}$ can better approximate the real target distribution and finally achieve enhanced performance.

## 4 Experiments

### 4.1 Experimental Settings

**Datasets.** We adopt four standard domain adaptation benchmarks of different scales for evaluations, including the small-scale Office-31 dataset [33], the medium-scale Office-Home dataset [41], and two large-scale datasets (*i.e.*, VisDA [28] and DomainNet-126 [27]). Refer to the supplementary material for complete dataset statistics and domain configurations.

**Comparative Methods.** We compare with 21 existing methods from three distinct groups: i) the baseline results from the source model, ii) **generation-based SFDA methods** CPGA [31], ASOGE [6], ISFDA [26], PS [9], DATUM [1], and DM-SFDA [4], and iii) **non-generation SFDA methods** including current state-of-the-art SFDA methods SHOT [20], NRC [50], GKD [36], HCL [13], AaD [49], AdaCon [2], CoWA [18], SCLM [39], ELR [53], PLUE [23], CRS [57], CPD [61], TPDS [35], DIFO [38], and ProDe [37].

**Implementation Details.** We employ stable-diffusion v1-5 [32] as the diffusion model to generate 512×512 images with 20 denoising steps. $\gamma_1 = 5.5$ in (1) and $\gamma_2 = 0$ in (6). We set the threshold $E$ to 0.01, and the total refinement iteration count $R$ to 10. Note that setting $E$ and $R$ to other values

Table 1: **Office-31** results (%) with ResNet-50. Methods with top three performance in each column are highlighted in red, orange, and yellow.

| Method | Venue | A→D | A→W | D→A | D→W | W→A | W→D | Avg. |
|---|---|---|---|---|---|---|---|---|
| | | Baseline method | | | | | | |
| Source | – | 79.7 | 77.6 | 65.5 | 97.9 | 63.8 | 99.8 | 80.7 |
| | | Generation-based method | | | | | | |
| CPGA [31] | IJCAI21 | 94.4 | 94.1 | 76.0 | 98.4 | 76.6 | 99.8 | 89.9 |
| ASOGE [6] | TCSVT23 | 95.6 | 94.1 | 74.3 | 98.1 | 74.2 | 99.7 | 89.3 |
| ISFDA [26] | CVPR24 | 95.3 | 94.2 | 76.4 | 98.3 | 77.5 | 99.9 | 90.3 |
| DM-SFDA [4] | - | 97.7 | 99.0 | 82.7 | 99.3 | 83.5 | 100.0 | 93.7 |
| | | None-generation method | | | | | | |
| SHOT [20] | ICML20 | 93.7 | 91.1 | 74.2 | 98.2 | 74.6 | 100. | 88.6 |
| NRC [50] | NIPS21 | 96.0 | 90.8 | 75.3 | 99.0 | 75.0 | 100. | 89.4 |
| GKD [36] | IROS21 | 94.6 | 91.6 | 75.1 | 98.7 | 75.1 | 100. | 89.2 |
| HCL [13] | NIPS21 | 94.7 | 92.5 | 75.9 | 98.2 | 77.7 | 100. | 89.8 |
| AaD [49] | NIPS22 | 96.4 | 92.1 | 75.0 | 99.1 | 76.5 | 100. | 89.9 |
| AdaCon [2] | CVPR22 | 87.7 | 83.1 | 73.7 | 91.3 | 77.6 | 72.8 | 81.0 |
| CoWA [18] | ICML22 | 94.4 | 95.2 | 76.2 | 98.5 | 77.6 | 99.8 | 90.3 |
| ELR [53] | ICLR23 | 93.8 | 93.3 | 76.2 | 98.0 | 76.9 | 100. | 89.6 |
| PLUE [23] | CVPR23 | 89.2 | 88.4 | 72.8 | 97.1 | 69.6 | 97.9 | 85.8 |
| CPD [61] | PR24 | 96.6 | 94.2 | 77.3 | 98.2 | 78.3 | 100. | 90.8 |
| TPDS [35] | IJCV24 | 97.1 | 94.5 | 75.7 | 98.7 | 75.5 | 99.8 | 90.2 |
| DIFO [49] | CVPR24 | 93.6 | 92.1 | 78.5 | 95.7 | 78.8 | 97.0 | 89.3 |
| ProDe [37] | ICLR25 | 94.4 | 92.1 | 79.8 | 95.6 | 79.0 | 98.6 | 89.9 |
| **DPTM(ours)** | – | 97.2 | 95.3 | 92.0 | 98.7 | 91.7 | 100. | 95.8 |

may obtain superior performance. For the adaptation model, we employ ResNet-50 for Office-31 [33], Office-Home [41] and DomainNet-126 [27], and ResNet-101 for VisDA [28]. We train for 20K iterations with the batch size of 128 and learning rate of 3$e$-3 for large-scale DomainNet-126 [27] and VisDA [28], and 15K iterations with the batch size of 32 and learning rate of 1$e$-3 for Office-31 and Office-Home. Weight decay is set to 5$e$-4 for all the datasets.

## 4.2 Main Results

**Evaluations on Office-31.** Table 1 shows that our method is superior to generation-based SFDA methods on Office-31 and outperforms the best generation-based SFDA method DM-SFDA [4] on average across all the DA tasks. Compared with non-generation methods, our method outperforms the best non-generation methods in all tasks except D→W, delivering an average accuracy gain of 5%. Notably, our method achieves significant improvements on challenging adaptation tasks: 9.3% on D→A and 8.2% on W→A. These results validate the effectiveness of our method.

**Evaluations on Office-Home and Visda.** Table 2 shows that our method significantly outperforms existing SFDA methods on Office-Home and VisDA. On Office-Home, we achieve an average accuracy gain of 11.7% over the best generation-based SFDA method DM-SFDA [4] and 10.1% over the current SOTA method ProDe [37] across all domain adaptation tasks. Remarkably, our method outperforms ProDe by 22.7%, 21.0%, and 21.6% on challenging Ar→Cl, Pr→Cl, and Rw→Cl tasks where existing methods usually perform poorly. On VisDA, our method achieves an average accuracy gain of 8.5% over ISFDA [26] and 8.2% ProDe [37] (see the supplementary material for details). These results strongly validate the effectiveness of our method in difficult domain adaptation scenarios.

**Evaluations on DomainNet-126.** Our method achieves a 17.6% higher average accuracy than the generation-based CPGA [31] and surpasses current SOTA ProDe [37] by 3.7% on DomainNet-126. It significantly outperforms CPGA [31] across all domain adaptation tasks and exceeds ProDe [37] in most tasks, with only minor performance gaps in three DA scenarios.

## 4.3 Ablation Studies

We conduct ablation studies mainly on the Office-Home dataset. More ablation studies can be found in the supplementary materials.

Table 2: Results (%) on **Office-Home** and **VisDA**. **Office-Home** is evaluated with ResNet-50, and **VisDA** is evaluated with ResNet-101. The top three performances in each column are highlighted in red, orange, and yellow, respectively.

| Method | Venue | Office-Home | | | | | | | | | | | | | VisDA |
|---|---|---|---|---|---|---|---|---|---|---|---|---|---|---|---|
| | | Ar→Cl | Ar→Pr | Ar→Rw | Cl→Ar | Cl→Pr | Cl→Rw | Pr→Ar | Pr→Cl | Pr→Rw | Rw→Ar | Rw→Cl | Rw→Pr | Avg. | Sy→Re |
| Baseline method | | | | | | | | | | | | | | | |
| Source | – | 50.1 | 67.9 | 74.4 | 55.2 | 65.2 | 67.2 | 53.4 | 44.5 | 74.1 | 64.2 | 51.5 | 78.7 | 62.2 | 63.5 |
| Generation-based method | | | | | | | | | | | | | | | |
| CPGA [31] | IJCAI21 | 59.3 | 78.1 | 79.8 | 65.4 | 75.5 | 76.4 | 65.7 | 58.0 | 81.0 | 72.0 | 64.4 | 83.3 | 71.6 | 86.0 |
| ASOGE [6] | TCSVT23 | 59.1 | 78.4 | 81.0 | 67.7 | 78.4 | 77.5 | 65.8 | 57.2 | 80.2 | 72.7 | 60.7 | 83.3 | 71.8 | 83.2 |
| ISFDA [26] | CVPR24 | 60.7 | 78.9 | 82.0 | 69.9 | 79.5 | 79.7 | 67.1 | 58.8 | 82.3 | 74.2 | 61.3 | 86.4 | 73.4 | 88.4 |
| PS [9] | ML24 | 57.8 | 77.3 | 81.2 | 68.4 | 76.9 | 78.1 | 67.8 | 57.3 | 82.1 | 75.2 | 59.1 | 83.4 | 72.1 | 84.1 |
| DATUM [1] | CVPR23 | 55.3 | 76.8 | 79.3 | 65.1 | 77.7 | 78.6 | 62.4 | 52.1 | 79.7 | 66.6 | 55.9 | 80.5 | 69.2 | – |
| DM-SFDA [4] | – | 68.5 | 89.6 | 83.3 | 70.0 | 85.8 | 87.4 | 71.3 | 69.6 | 88.2 | 77.8 | 68.5 | 88.7 | 79.5 | 86.3 |
| None-generation method | | | | | | | | | | | | | | | |
| SHOT [20] | ICML20 | 56.7 | 77.9 | 80.6 | 68.0 | 78.0 | 79.4 | 67.9 | 54.5 | 82.3 | 74.2 | 58.6 | 84.5 | 71.9 | 82.7 |
| NRC [50] | NIPS21 | 57.7 | 80.3 | 82.0 | 68.1 | 79.8 | 78.6 | 65.3 | 56.4 | 83.0 | 71.0 | 58.6 | 85.6 | 72.2 | 85.9 |
| GKD [36] | IROS21 | 56.5 | 78.2 | 81.8 | 68.7 | 78.9 | 79.1 | 67.6 | 54.8 | 82.6 | 74.4 | 58.5 | 84.8 | 72.2 | 83.0 |
| AaD [49] | NIPS22 | 59.3 | 79.3 | 82.1 | 68.9 | 79.8 | 79.5 | 67.2 | 57.4 | 83.1 | 72.1 | 58.5 | 85.4 | 72.7 | 88.0 |
| AdaCon [2] | CVPR22 | 47.2 | 75.1 | 75.5 | 60.7 | 73.3 | 73.2 | 60.2 | 45.2 | 76.6 | 65.6 | 48.3 | 79.1 | 65.0 | 86.8 |
| CoWA [18] | ICML22 | 56.9 | 78.4 | 81.0 | 69.1 | 80.0 | 79.9 | 67.7 | 57.2 | 82.4 | 72.8 | 60.5 | 84.5 | 72.5 | 86.9 |
| SCLM [39] | NN22 | 58.2 | 80.3 | 81.5 | 69.3 | 79.0 | 80.7 | 69.0 | 56.8 | 82.7 | 74.7 | 60.6 | 85.0 | 73.0 | 85.3 |
| ELR [53] | ICLR23 | 58.4 | 78.7 | 81.5 | 69.2 | 79.5 | 79.3 | 66.3 | 58.0 | 82.6 | 73.4 | 59.8 | 85.1 | 72.6 | 85.8 |
| PLUE [23] | CVPR23 | 49.1 | 73.5 | 78.2 | 62.9 | 73.5 | 74.5 | 62.2 | 48.3 | 78.6 | 68.6 | 51.8 | 81.5 | 66.9 | 88.3 |
| CPD [61] | PR24 | 59.1 | 79.0 | 82.4 | 68.5 | 79.7 | 79.5 | 67.9 | 57.9 | 82.8 | 73.8 | 61.2 | 84.6 | 73.0 | 85.8 |
| TPDS [35] | IJCV24 | 59.3 | 80.3 | 82.1 | 70.6 | 79.4 | 80.9 | 69.8 | 56.8 | 82.1 | 74.5 | 61.2 | 85.3 | 73.5 | 87.6 |
| DIFO [38] | CVPR24 | 62.6 | 87.5 | 87.1 | 79.5 | 87.9 | 87.4 | 78.3 | 63.4 | 88.1 | 80.0 | 63.3 | 87.7 | 79.4 | 88.6 |
| ProDe [37] | ICLR25 | 64.0 | 90.0 | 88.3 | 81.1 | 90.1 | 88.6 | 79.8 | 65.4 | 89.0 | 80.9 | 65.5 | 90.2 | 81.1 | 88.7 |
| **DPTM(ours)** | – | 86.7 | 94.2 | 92.8 | 91.5 | 94.0 | 92.6 | 90.6 | 86.4 | 92.8 | 90.5 | 87.1 | 94.7 | 91.2 | 97.6 |

Table 3: Results (%) on **DomainNet-126** evaluated with ResNet-50. The top three performances in each column are highlighted in red, orange, and yellow, respectively.

| Method | Venue | DomainNet-126 | | | | | | | | | | | | |
|---|---|---|---|---|---|---|---|---|---|---|---|---|---|---|
| | | C→P | C→R | C→S | P→C | P→R | P→S | R→C | R→P | R→S | S→C | S→P | S→R | Avg. |
| Baseline method | | | | | | | | | | | | | | |
| Source | – | 47.5 | 59.8 | 48.6 | 51.0 | 75.3 | 47.8 | 57.5 | 61.1 | 48.6 | 63.5 | 56.2 | 59.5 | 56.4 |
| Generation-based method | | | | | | | | | | | | | | |
| CPGA [31] | IJCAI21 | 61.2 | 76.7 | 59.6 | 64.5 | 81.3 | 61.0 | 68.6 | 69.5 | 65.9 | 66.9 | 60.2 | 75.1 | 67.6 |
| None-generation method | | | | | | | | | | | | | | |
| SHOT [20] | ICML20 | 63.5 | 78.2 | 59.5 | 67.9 | 81.3 | 61.7 | 67.7 | 67.6 | 57.8 | 70.2 | 64.0 | 78.0 | 68.1 |
| NRC [50] | NIPS21 | 62.6 | 77.1 | 58.3 | 62.9 | 81.3 | 60.7 | 64.7 | 69.4 | 58.7 | 69.4 | 65.8 | 78.7 | 67.5 |
| GKD [36] | IROS21 | 61.4 | 77.4 | 60.3 | 69.6 | 81.4 | 63.2 | 68.3 | 68.4 | 59.5 | 71.5 | 65.2 | 77.6 | 68.7 |
| AdaCon [2] | CVPR22 | 60.8 | 74.8 | 55.9 | 62.2 | 78.3 | 58.2 | 63.1 | 68.1 | 55.6 | 67.1 | 66.0 | 75.4 | 65.4 |
| CoWA [18] | ICML22 | 64.6 | 80.6 | 60.6 | 66.2 | 79.8 | 60.8 | 69.0 | 67.2 | 60.0 | 69.0 | 65.8 | 79.9 | 68.6 |
| PLUE [23] | CVPR23 | 59.8 | 74.0 | 56.0 | 61.6 | 78.5 | 57.9 | 61.6 | 65.9 | 53.8 | 67.5 | 64.3 | 76.0 | 64.7 |
| TPDS [35] | IJCV24 | 62.9 | 77.1 | 59.8 | 65.6 | 79.0 | 61.5 | 66.4 | 67.0 | 58.2 | 68.6 | 64.3 | 75.3 | 67.1 |
| DIFO [38] | CVPR24 | 73.8 | 89.0 | 69.4 | 74.0 | 88.7 | 70.1 | 74.8 | 74.6 | 69.6 | 74.7 | 74.3 | 88.0 | 76.7 |
| ProDe [37] | ICLR25 | 79.3 | 91.0 | 75.3 | 80.0 | 90.9 | 75.6 | 80.4 | 78.9 | 75.4 | 80.4 | 79.2 | 91.0 | 81.5 |
| **DPTM(ours)** | – | 85.6 | 90.9 | 80.0 | 85.1 | 90.7 | 79.0 | 85.2 | 85.4 | 78.1 | 86.1 | 85.4 | 90.9 | 85.2 |

Table 4: Ablation study results (%) on Different LDMs evaluated with $R = 3$, $E = 0.001$.

| LDM | Ar→Cl | Ar→Pr | Ar→Rw | Cl→Ar | Cl→Pr | Cl→Rw | Pr→Ar | Pr→Cl | Pr→Rw | Rw→Ar | Rw→Cl | Rw→Pr | Avg. |
|---|---|---|---|---|---|---|---|---|---|---|---|---|---|
| SDXL | **69.4** | 82.2 | 82.8 | 69.4 | 82.6 | 82.2 | **65.8** | **67.0** | 82.4 | **71.1** | **67.5** | 84.2 | **75.6** |
| SD15 | 67.0 | **83.6** | **83.9** | **70.6** | **84.3** | **82.9** | 65.6 | 63.6 | **84.6** | 69.6 | 66.4 | **85.0** | **75.6** |

**Different Versions of Latent Diffusion Models.** We conduct ablation studies using both Stable Diffusion v1.5 (SD15) and Stable Diffusion XL (SDXL) [29], with identical parameters ($E = 0.001$ and $R = 3$) except for output resolution - SDXL natively generates 1024×1024 images while SD15 pro-

Table 5: Ablation study results (%) on Threshold $E$ evaluated with $R = 10$.

| $E$ | Ar→Cl | Ar→Pr | Ar→Rw | Cl→Ar | Cl→Pr | Cl→Rw | Pr→Ar | Pr→Cl | Pr→Rw | Rw→Ar | Rw→Cl | Rw→Pr | Avg. |
|---|---|---|---|---|---|---|---|---|---|---|---|---|---|
| 0.001 | 74.7 | 85.9 | 87.5 | 75.0 | 88.5 | 85.6 | 72.1 | 73.3 | 87.2 | 75.2 | 74.6 | 88.8 | 80.7 |
| 0.005 | 81.7 | 92.6 | 90.8 | 85.5 | 92.1 | 88.2 | 85.5 | 81.2 | 88.3 | 82.1 | 79.9 | 92.1 | 86.7 |
| 0.01 | **86.7** | **94.2** | **92.8** | **91.5** | **94.0** | **92.6** | **90.6** | **86.4** | **92.8** | **90.5** | **87.1** | **94.7** | **91.2** |

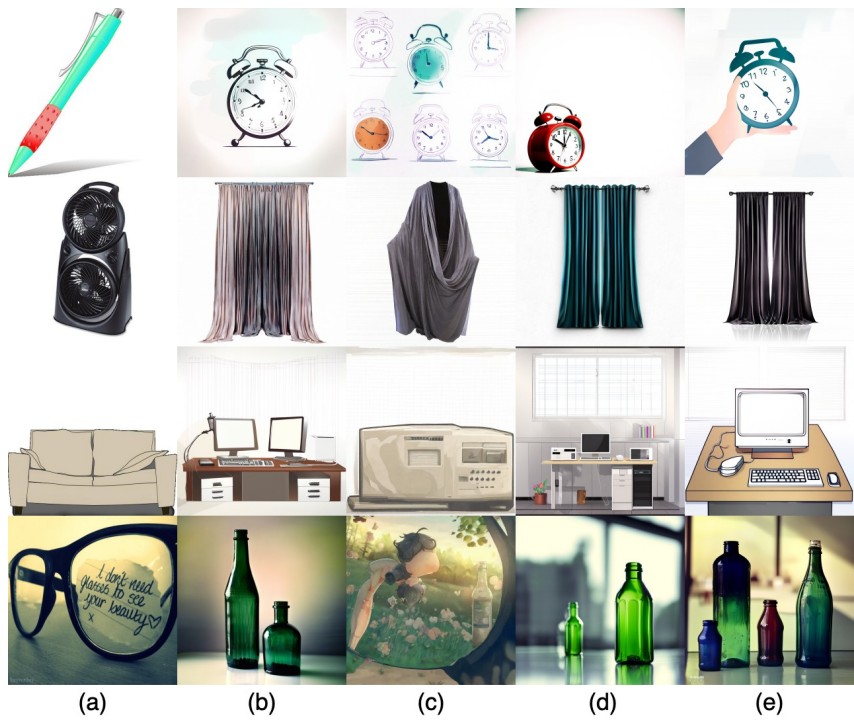

(a)  (b)  (c)  (d)  (e)

Figure 2: Ablation on Manipulation Mechanism of $\mathbf{x}_l^u$. Row: $\hat{y}_l$ = 'Alarm Clock', 'Curtains', 'Computer', 'Bottle', respectively. Column: (a) $\mathbf{x}_l^u$ (b) $\tilde{\mathbf{x}}_l^u$ w/o Target-guided Initialization (c) $\tilde{\mathbf{x}}_l^u$ w/o Semantic Feature Injection (d) $\tilde{\mathbf{x}}_l^u$ w/o Domain-specific Feature Preservation (e) $\tilde{\mathbf{x}}_l^u$ of our method.

duces $512\times512$ images due to their architectural differences. Table 4 shows comparable performance, but SDXL's higher computational cost makes SD15 our preferred choice for implementation.

**Values of Threshold $E$.** We also conduct ablation on the Threshold $E = \{0.001, 0.005, 0.01\}$. Table 5 shows that appropriately increasing $E$ may yield better results.

**Manipulation Mechanism of Non-trust Set.** As shown in Figure 2, our method's manipulated samples $\tilde{\mathbf{x}}_l^u$ exhibit the best semantic alignment with their assigned labels $\hat{y}_l = $ and the best preservation of target distribution characteristics, detailed in the supplementary material.

### 4.4 Additional Analysis

We provide additional analyses to further validate the effectiveness of our method. More analysis can be found in the supplementary materials.

**Analysis on Progressive Refinement Mechanism.** For the performance of our method on the Office-Home dataset shown in Table 2, we provide a detailed performance trajectory as $r$ increases from 1 to 10, in order to demonstrate the effectiveness of the proposed Progressive Refinement Mechanism. **Firstly**, we present experimental results for $r = \{0, 2, 4, 6, 8, 10\}$ in Table 6 and provide complete experimental results in the supplementary materials, where $r = 0$ is equivalent to using only the source model. Table 6 shows that the performance of the target model improves as $r$ increases. Specifically, as $r$ increases, the performance of the target model first improves rapidly and then growth becomes slow. **Secondly**, we select the first 4 DA tasks Ar→Cl, Ar→Pr, Ar→Rw, and Cl→Ar, and

Table 6: Full results of the performance trajectory as $r$ grows from 1 to 10 on Office-Home evaluated with $E = 0.01$.

| $r$ | Ar→Cl | Ar→Pr | Ar→Rw | Cl→Ar | Cl→Pr | Cl→Rw | Pr→Ar | Pr→Cl | Pr→Rw | Rw→Ar | Rw→Cl | Rw→Pr | Avg. |
|---|---|---|---|---|---|---|---|---|---|---|---|---|---|
| 0 | 50.1 | 67.9 | 74.4 | 55.2 | 65.2 | 67.2 | 53.4 | 44.5 | 74.1 | 64.2 | 51.5 | 78.7 | 62.2 |
| 1 | 60.2 | 80.9 | 82.8 | 67.8 | 79.3 | 80.4 | 64.5 | 57.1 | 81.6 | 69.1 | 60.5 | 83.4 | 72.3 |
| 2 | 72.6 | 87.2 | 85.7 | 73.5 | 85.4 | 84.4 | 72.3 | 69.7 | 85.2 | 76.7 | 70.8 | 87.7 | 79.3 |
| 3 | 75.3 | 89.7 | 87.6 | 77.7 | 88.9 | 86.4 | 79.2 | 76.1 | 86.7 | 80.1 | 75.5 | 90.2 | 82.8 |
| 4 | 79.0 | 91.8 | 89.1 | 81.0 | 91.2 | 88.3 | 82.1 | 79.1 | 88.1 | 82.7 | 79.1 | 91.7 | 85.3 |
| 5 | 81.5 | 92.5 | 89.8 | 83.9 | 92.0 | 89.4 | 85.3 | 81.5 | 89.3 | 84.5 | 81.8 | 92.8 | 87.0 |
| 6 | 83.7 | 92.9 | 90.7 | 86.1 | 92.8 | 90.2 | 87.1 | 83.1 | 90.0 | 87.4 | 83.3 | 93.3 | 88.4 |
| 7 | 85.0 | 93.7 | 91.5 | 87.5 | 93.0 | 91.2 | 87.8 | 84.4 | 91.0 | 88.6 | 84.9 | 93.9 | 89.4 |
| 8 | 85.9 | 94.2 | 91.7 | 88.7 | 93.8 | 91.9 | 89.0 | 85.1 | 91.7 | 89.7 | 85.6 | 94.2 | 90.1 |
| 9 | 85.9 | 94.1 | 92.0 | 89.8 | 93.9 | 92.3 | 89.8 | 85.6 | 92.2 | 90.2 | 86.6 | 94.4 | 90.6 |
| **10** | **86.7** | **94.2** | **92.8** | **91.5** | **94.0** | **92.6** | **90.6** | **86.4** | **92.8** | **90.5** | **87.1** | **94.7** | **91.2** |

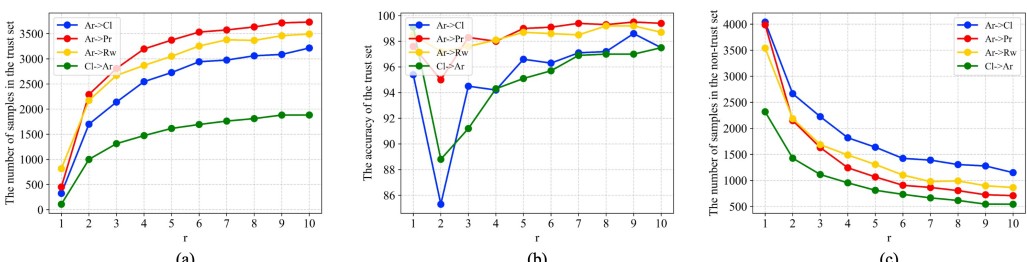

Figure 3: The relationship between $r$ versus: (a) The number of samples in the trust set. (b) The trust set accuracy. (c) The number of samples in the non-trust set.

plot: the relationship between the number of samples in the trust set versus $r$, the trust set accuracy versus $r$, and the number of samples in the non-trust set versus $r$. As shown in Figure 3, two main conclusions can be drawn: (1) The number of samples in the trust set increases significantly with $r$, while correspondingly, the non-trust set size decreases substantially. This indicates that the model progressively learns to make predictions with low uncertainty. (2) Overall, the trust set accuracy remains at a high level. Although relatively low across all four DA tasks at $r = 2$, the accuracy shows significant recovery with increasing $r$, demonstrating our method's capability to progressively correct previous errors.

**Analysis on Trust and Non-trust Partition for Target Domain.** We provide more analysis on the proposed Trust and Non-trust Partition for the target Domain, detailed in the supplementary materials.

**Analysis on Manipulation of Non-trust Set.** We provide more analysis on the proposed Manipulation of Non-trust Set, detailed in the supplementary materials.

## 5 Conclusion

We propose DPTM, a novel generation-based framework that utilizes unlabeled target data as references to construct and progressively refine a pseudo-target domain via the latent diffusion model for Source-free Domain Adaptation (SFDA). We first divide the target into a trust set and a non-trust set based on prediction uncertainty. For the trust set, we directly train the target model with pseudo labels in a supervised manner. For the non-trust set, we assign a label for each sample and propose a manipulation strategy consisting of Target-guided Initialization, Semantic Feature Injection, and Domain-specific Feature Preservation, which semantically transforms the high-uncertainty sample toward the assigned category, while maintaining the generated sample in the target distribution. We progressively refined this process which simultaneously corrects pseudo-label inaccuracies in the previous trust set and decreases domain discrepancy in the previous pseudo-target domain, iteratively improving the target model. Experimental results demonstrate that our method achieves state-of-the-art performance on SFDA classification.

## Acknowledgements

This work was supported in part by the National Natural Science Foundation of China under Grant 62431017, Grant 62320106003, Grant U24A20251, Grant 62125109, Grant 62120106007, Grant 62371288, Grant 62301299, Grant 62401366, Grant 62401357, Grant 62401367, and in part by the Program of Shanghai Science and Technology Innovation Project under Grant 24BC3200800.

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
