# OpenReview forum: "Diffusion-Driven Progressive Target Manipulation for Source-Free Domain Adaptation"
_NeurIPS.cc/2025/Conference — NeurIPS 2025 poster_

### Official Review · Reviewer_mAks · 2025-06-26

**Clarity:** 3
**Significance:** 2
**Originality:** 3
**Rating:** 4
**Confidence:** 4

**Summary:**

This paper proposes a novel approach to the unsupervised source-free domain adaptation problem by leveraging diffusion models to transform unlabeled target images into images of specified categories. Specifically, the method first separates given target samples into a trust set and a non-trust set based on predictive uncertainty. For the samples in the non-trust set, a diffusion model is used to transform them into labeled images while preserving the characteristics of the target domain. By iteratively repeating this process, the authors demonstrate that their method achieves strong domain adaptation performance even in scenarios with a large source-target domain gap.

**Questions:**

* The diffusion process relies on the assumption that for a sufficiently large $T$, the latent variable $z_T$ follows a Gaussian distribution. However, Eq. (5) assumes that low-frequency information remains in $z_T$. I am curious whether there are specific conditions under which this assumption holds.
* In the progressive refinement mechanism described in Section 3.4, the manipulation is repeatedly applied to all samples in $U^{r+1}$. However, this manipulation process appears to be independent of the target model $\phi^r_{trg}$, then already manipulated samples can be reused. Would it not be more reasonable to apply the manipulation only to the newly added samples in $U^{r+1}$? Does this design choice have any impact on the overall performance?
* The proposed method is based on the assumption that domain-specific and semantic information can be separated via FFT. While this may reasonably hold for the benchmark datasets used in the paper, it is unclear whether the same assumption would remain valid for real-image benchmarks that are commonly used in SFDA, such as TerraIncognita or VLCS.

**Ethical Concerns:**

["NO or VERY MINOR ethics concerns only"]

**Final Justification:**

The experimental and analytical results, along with the detailed explanations provided by the authors during the rebuttal process, have addressed many of my concerns about the paper. I hope that the new results presented by the authors will be properly incorporated into the revised version. In particular, I believe that the results on unsupervised model selection are valuable to support the practical applicability of the method.

**Limitations:**

* The authors appropriately acknowledge the limitation that their method relies on a pretrained diffusion model. However, as noted in the weaknesses section, it is also important to discuss the limitations concerning the practical applicability of the proposed approach. In particular, I am concerned that without a clearly defined model selection procedure under the unsupervised SFDA setting, the method may yield results that do not generalize beyond the specific benchmarks used.

**Paper Formatting Concerns:**

No formatting issues

**Quality:**

3

**Strengths And Weaknesses:**

* Strengths
  * This work explores a novel approach that leverages the powerful generative capability of pretrained diffusion models to convert given unlabeled target samples into labeled target samples. To preserve target-domain-specific features during this transformation, the authors propose several FFT-based techniques.
  * The paper evaluates the proposed method across a range of benchmark datasets and compares it with various existing unsupervised SFDA approaches, focusing on scenarios with significant domain shifts.

* Weaknesses
  * The most significant concern with this work lies in its practical applicability to real-world scenarios. The proposed method involves several hyperparameters, such as the uncertainty threshold $E$, FFT-related parameters, and guidance scales $\gamma_1$ and $\gamma_2$, but the paper does not discuss how these values are determined. By definition, in unsupervised SFDA, these hyperparameters must be chosen using only the unlabeled target samples. However, the paper omits any discussion of this critical issue. While prior works have often reported results using hyperparameters tuned based on target-domain performance, I believe this convention should not be followed, as it entirely neglects the practical usability and generalizability of the method. There is a growing body of literature addressing hyperparameter selection in the context of SFDA, and at the very least, the authors should evaluate whether their method remains effective when such methods are employed. Relevant references include:
    * Saito et al., “Tune it the right way: Unsupervised validation of domain adaptation via soft neighborhood density,” ICCV 2021
    * Hu et al., “Mixed samples as probes for unsupervised model selection in domain adaptation,” NeurIPS 2023
    * Lee et al., “Few-shot fine-tuning is all you need for source-free domain adaptation,” arXiv:2304.00792
  * I also have concerns regarding the reported effectiveness of the proposed method. Related to the above point, the ablation study on the uncertainty threshold $E$ clearly shows how sensitive the performance is to hyperparameter choices. For instance, Table 6 shows that when $E=0.001$, the average performance drops to 80.7, at which point the method no longer shows meaningful improvements over existing approaches. Additionally, in the performance comparisons, results from baselines such as ProDe appear selectively chosen. ProDe, for example, reports significantly higher performance when using a ViT-CLIP backbone, which is not shown in this paper. This undermines the objectivity of the evaluation and should be avoided.
  * The proposed method involves several complex components that serve similar functions, yet the effectiveness of this design is only qualitatively illustrated (e.g., in Figure 3). A more thorough, quantitative component-wise analysis is necessary to provide a clearer understanding of the contribution of each part of the method.

---

> ### Author Rebuttal · Authors · 2025-07-31
>
> ### Q1: Hyperparameter selection procedure and real-world scenarios performance.
> We thank the reviewer’s insightful comments regarding the practical applicability of our method in real-world scenarios, particularly in the hyperparameter selection procedure. We respond to this concern from two aspects:
> #### 1. Hyperparameter determination.
> We would like to clarify the determination of hyperparameters with a more detailed description. In fact, we did not tune hyperparameters separately for each SFDA task. The guidance scales and FFT-related hyperparameters are fixed on all the datasets, and we also use $E=0.01$ and $R=10$ for all datasets. In lines 254-255, we intend to mean that we only explore different values of $E$ and $R$ once on Office-Home and fix $E$ to 0.01 and $R$ to 10 for all the experiments (except for ablation studies). Thus, the hyperparameter selection is practical for applications in real-world scenarios. We will further discuss the practicality of this hyperparameter below and validate it on VLCS.
>
> #### 2. Recommendations for real-world usage.
> We fully agree that hyperparameter determination can be hard in real-world scenarios. Thus, we provide practical recommendations for hyperparameter selection.
> * For **default settings**, we suggest using the same hyperparameters as we do.
> * For **adjustment strategy**, if the model does not perform well, we first recommend moderately increasing both $E$ and $R$ (*e.g.*, $E=0.1$ and $R=20$). Increasing $E$ and $R$ is probably effective in most cases, according to Section 4.3.
> * For more complex scenarios, we can also explore other hyperparameter values. In such cases, **unsupervised model selection** becomes necessary. Inspired by the reviewer, we employ an unsupervised model selection strategy using nuclear norm. We extract the softmax output matrix for all target samples and calculate its nuclear norms. Then， we select the model with the largest nuclear norm [1][2]. As suggested, we train models with $E\in[0.01,0.005,0.001]$ and $R$ from 1 to 10 on real-world benchmark **VLCS** and apply this model selection metric. In the table below,  comparison results are from [3], where UMS is **unsupervised model selection** for short. Note that all the methods for comparison report their best target accuracy (according to the labels), **while ours performs model selection without knowing target accuracy**.
>
> | Method | UMS | C→L | C→S | C→V | L→C | L→S | L→V | S→C | S→L | S→V | V→C | V→L | V→S | Avg |
> |--------|-----|-----|-----|-----|-----|-----|-----|-----|-----|-----|-----|-----|-----|-----|
> | No adapt | ✗ | 47.8 | 53.3 | 64.6 | 51.3 | 40.7 | 55.1 | 58.1 | 36.4 | 55.4 | 97.8 | 48.8 | 72.0 | 56.7 |
> | SHOT [28] | ✗ | 44.6 | 55.7 | 75.9 | 65.3 | 60.8 | 74.8 | 87.7 | 41.1 | 82.7 | 89.8 | 45.8 | 65.7 | 65.6 |
> | SHOT++ [30] | ✗ | 41.7 | 58.4 | 75.6 | 70.0 | 56.9 | 76.2 | 71.6 | 39.5 | 80.7 | 96.9 | 43.7 | 60.07 | 64.3 |
> | AaD [59] | ✗ | 37.7 | 57.2 | 75.5 | 59.5 | 48.8 | 67.5 | 84.0 | 34.8 | 72.8 | 43.4 | 40.2 | 54.8 | 56.4 |
> | CoWA-JMDS [23] | ✗ | 46.1 | 58.4 | 81.1 | 85.6 | 64.1 | 78.2 | 95.9 | 48.1 | 82.7 | 99.4 | 50.8 | 65.3 | 71.3 |
> | NRC [37] | ✗ | 39.9 | 55.9 | 75.5 | 64.7 | 54.1 | 74.8 | 78.4 | 40.3 | 82.2 | 90.1 | 41.7 | 62.8 | 63.4 |
> | G-SFDA [58] | ✗ | 42.6 | 54.9 | 73.5 | 82.3 | 51.4 | 72.0 | 74.6 | 45.8 | 82.7 | 88.9 | 49.1 | 64.3 | 65.2 |
> | FT | ✗ | 50.1 | 66.7 | 81.1 | 99.7 | 62.2 | 78.0 | 99.8 | 55.5 | 80.5 | 99.7 | 53.0 | 67.0 | 74.5 |
> | LP-FT | ✗ | 50.5 | 66.7 | 79.5 | 99.7 | 65.6 | 77.6 | 99.7 | 54.1 | 79.5 | 99.7 | 51.2 | 69.8 | 74.5 |
> | **Ours** | ✓ | 88.2 | 96.0 | 90.8 | 97.7 | 97.2 | 99.2 | 99.6 | 88.6 | 89.4 | 99.9 | 92.7 | 91.5 | 94.2 |
>
> [1]Cui S et al. Towards discriminability and diversity: Batch nuclear-norm maximization under label insufficient situations, CVPR 2020.
> [2]Cui S et al. Fast batch nuclear-norm maximization and minimization for robust domain adaptation. arXiv:2107.06154.
> [3]Lee S et al. Few-shot fine-tuning is all you need for source-free domain adaptation. arXiv:2304.00792.
>
> ### Q2：$E$ sensitivity.
> As mentioned above, we uniformly set $E=0.01$ for all experiments across datasets. The ablation results with $E=0.001$ are only provided to demonstrate that a higher $E$ yields better performance, and this setting ($E=0.001$) was never actually used in Tables 1–3.
>
> ### Q3: Selectively chosen of baselines in the performance comparisons.
> In the ProDe and DIFO papers, ProDe-V and DIFO-C-B32 indicate direct use of CLIP-ViT as the backbone for the SFDA model. Nevertheless, most SFDA methods, including ours, use vanilla ResNet as the backbone. For fair comparison, we did not include ProDe-V and DIFO-C-B32. We have clearly specified the ResNet backbone in the captions of Tables 1-3.
>
> ### Q4: Quantitative component-wise analysis.
>
> Ablation studies are performed on each component in Response to Q1 by Reviewer fShy, and demonstrate the effectiveness of each component.
>
> ### Q5: Low-frequency information remains in diffusion latents.
> Yes. Prior studies [4][5][6] have demonstrated this conclusion, and they also operated frequency decomposition in the diffusion latent space.
>
> ### Q6: Reuse already manipulated samples.
> The target model determines which samples are included in the non-trust set, and usually selects those it struggles to classify. After training on their manipulated version, some non-trust samples become well-understood, as implied by the growing trust set. Therefore, retaining their manipulated versions is unnecessary and introduces two issues:
> * It creates an ever-growing synthetic domain which is much larger than the real target domain, causing the model to adapt to the synthetic domain rather than the real target domain.
> * It significantly increases training costs.
> We offer a similar experimental analysis in Response to Q4 by Reviewer pUPg.
>
> [4]Everaert et al. Exploiting the signal-leak bias in diffusion models. WACV 2024.
>
> [5]Wu et al. Freeinit: Bridging initialization gap in video diffusion models. ECCV 2024.
>
> [6]Koo G et al. Flexiedit: Frequency-aware latent refinement for enhanced non-rigid editing. ECCV 2024.

---

> > ### Comment · Reviewer_mAks · 2025-08-04
> >
> > Thank you to the authors for their thoughtful and thorough responses. Most of my concerns have been addressed through the rebuttal. I hope the following points can be incorporated into the revised version of the paper.
> >
> > As I previously noted, the question of how to select the optimal model under an unsupervised scenario deserves careful discussion. I appreciate the authors' effort in presenting hyperparameter selection results using the nuclear norm on the VLCS dataset, and I encourage the inclusion of this result in the updated version. While the nuclear norm may not be the optimal criterion, it is nonetheless meaningful in that it reflects the constraints of an unsupervised setting. Applying similar unsupervised validation criteria in other experimental settings could further strengthen the practical value of the work.
> >
> > In addition, I hope the authors include the ablation study results for the proposed components. Quantitative evaluation of each component’s contribution is essential for understanding the effectiveness of the overall method.

---

> ### Author Response · Authors · 2025-08-07
> **New results 1: Unsupervised model selection**
>
> We sincerely thank the reviewer for the constructive and insightful feedback. We fully agree with the importance of both aspects raised, and we will incorporate these points into the revised version of our manuscript. In addition, we have further extended our evaluation along both aspects and present the corresponding results below.
>
> ## New results 1: Unsupervised model selection
>
> The reviewer previously suggested evaluating the performance of our method on real-world scenarios, including VLCS and TerraIncognita, to further validate the practicality of our method. In our initial response, due to time constraints, we only conducted experiments on the VLCS dataset.
>
> We have now additionally conducted **unsupervised model selection** experiments on the **TerraIncognita dataset** to further evaluate the practicality of our method. The experimental setting follows that of the VLCS dataset, and we also use the nuclear norm to select the best model without access to target domain labels. The comparison results are also taken from [1].
>
> | Method          | UMS | L100→L38 | L100→L43 | L100→L46 | L38→L100 | L38→L43 | L38→L46 | L43→L100 | L43→L38 | L43→L46 | L46→L100 | L46→L38 | L46→L43 |  Avg |
> |-----------------|-----|:--------:|:--------:|:--------:|:--------:|:-------:|:-------:|:--------:|:-------:|:-------:|:--------:|:-------:|:-------:|:----:|
> | No adapt        |  ✗  |   26.2   |   20.3   |   27.1   |   29.3   |   31.4  |   31.6  |   24.1   |   44.1  |   38.7  |   33.6   |   21.6  |   22.2  | 29.2 |
> | SHOT            |  ✗  |   20.1   |   23.8   |   28.5   |   36.0   |   29.0  |   13.6  |   26.2   |   14.5  |   32.7  |   34.3   |   12.6  |   37.4  | 25.7 |
> | SHOT++          |  ✗  |   29.3   |   22.1   |   25.5   |   22.8   |   31.8  |   18.4  |   33.3   |   22.6  |   25.6  |   35.8   |   13.0  |   44.6  | 27.1 |
> | AaD             |  ✗  |   17.2   |   17.4   |   22.1   |   24.6   |   28.1  |   13.3  |   28.9   |   23.3  |   23.1  |   31.6   |   7.4   |   34.6  | 22.6 |
> | CoWA-JMDS       |  ✗  |   33.1   |   31.4   |   26.4   |   36.3   |   38.3  |   19.3  |   28.2   |   13.6  |   26.6  |   32.5   |   10.0  |   47.6  | 28.7 |
> | NRC             |  ✗  |   19.3   |   22.7   |   29.5   |   38.5   |   26.9  |   14.9  |   30.8   |   22.6  |   32.2  |   28.9   |   11.0  |   39.0  | 26.4 |
> | G-SFDA          |  ✗  |   21.6   |   29.1   |   38.2   |   38.4   |   27.0  |   22.4  |   40.9   |   17.4  |   33.3  |   35.0   |   16.3  |   52.6  | 31.0 |
> | FT              |  ✗  |   52.4   |   41.7   |   50.0   |   63.8   |   38.6  |   47.8  |   66.2   |   56.7  |   51.4  |   68.9   |   56.7  |   61.2  | 54.6 |
> | LP-FT           |  ✗  |   54.3   |   47.5   |   46.9   |   63.6   |   41.3  |   49.0  |   64.2   |   55.9  |   54.4  |   68.4   |   55.7  |   63.8  | 55.4 |
> | **Ours (DPTM)** |  ✓  |   61.3   |   82.3   |   58.3   |   74.7   |   77.8  |   50.2  |   69.1   |   66.1  |   58.3  |   75.1   |   63.6  |   77.0  | 67.8 |
>
> The results demonstrate that, similar to VLCS, our method significantly outperforms all compared methods even when using nuclear norm as the criterion for **unsupervised model selection**, achieving an average accuracy improvement of **12.4%** over the best-performing comparative method. Notably, in some **challenging scenarios** such as L100→L43 and L38→L43, where existing methods generally perform poorly, our method achieves substantial gains of **34.8% and 36.5%**, respectively, even **under unsupervised model selection**. It is worth noting that all compared methods were evaluated using their best-performing models selected based on target domain accuracy. These results further demonstrate the effectiveness and practicality of our method.
>
> We also acknowledge that there may exist better criteria beyond the nuclear norm. Exploring more effective model selection metrics will be an important direction for our future work, as we believe that a more suitable criterion could further unleash the potential of our method and enhance its practical applicability. We sincerely thank the reviewer for the valuable insights.
>
> [1] Lee S et al. Few-shot fine-tuning is all you need for source-free domain adaptation. arXiv:2304.00792.

---

> ### Author Response · Authors · 2025-08-07
> **New results 2: Component-wise ablation study results**
>
> ## New results 2: Component-wise ablation study results
>
> We further conduct ablation studies on **the independent usage of each individual component**. For clarity in the tables, we refer to Target-guided Initialization, Semantic Feature Injection, and Domain-specific Feature Preservation as **TGI, SFI, and DFP**, respectively. We present comprehensive component-wise ablation results, including the performance of the model **with only one component enabled** and **with each component individually removed**.
>
> | TGI | SFI | DFP | Ar→Cl | Ar→Pr | Ar→Rw | Cl→Ar | Cl→Pr | Cl→Rw | Pr→Ar | Pr→Cl | Pr→Rw | Rw→Ar | Rw→Cl | Rw→Pr | Avg. |
> |-----|-----|-----|-------|-------|-------|-------|-------|-------|-------|-------|-------|-------|-------|-------|------|
> | ✗   | ✗   | ✗   | 50.1  | 67.9  | 74.4  | 55.2  | 65.2  | 67.2  | 53.4  | 44.5  | 74.1  | 64.2  | 51.5  | 78.7  | 62.2 |
> | ✓   | ✗   | ✗   | 69.2  | 86.2  | 82.2  | 74.6  | 87.8  | 80.7  | 76.7  | 67.8  | 82.4  | 73.5  | 66.9  | 87.1  | 77.9 |
> | ✗   | ✓   | ✗   | 59.6  | 80.8  | 82.3  | 67.9  | 83.1  | 80.2  | 64.6  | 68.1  | 81.7  | 70.6  | 67.9  | 86.5  | 74.4 |
> | ✗   | ✗   | ✓   | 67.8  | 87.6  | 82.7  | 68.7  | 84.0  | 80.6  | 70.7  | 67.6  | 82.4  | 74.1  | 67.1  | 88.7  | 76.8 |
> | ✗   | ✓   | ✓   | 75.9  | 89.4  | 88.3  | 78.9  | 87.9  | 87.0  | 76.0  | 74.1  | 87.8  | 80.0  | 75.9  | 89.1  | 82.5 |
> | ✓   | ✗   | ✓   | 72.9  | 88.0  | 84.1  | 77.5  | 88.3  | 83.6  | 76.8  | 69.0  | 84.0  | 75.7  | 68.8  | 88.4  | 79.8 |
> | ✓   | ✓   | ✗   | 70.2  | 89.9  | 85.5  | 81.1  | 89.4  | 89.3  | 80.7  | 70.7  | 86.3  | 80.3  | 72.1  | 91.0  | 82.2 |
> | ✓   | ✓   | ✓   | 86.7  | 94.2  | 92.8  | 91.5  | 94.0  | 92.6  | 90.6  | 86.4  | 92.8  | 90.5  | 87.1  | 94.7  | 91.2 |
>
>
> These results further demonstrate the effectiveness of our method. We provide a detailed analysis as below:
>
> For Target-guided Initialization (TGI for short):
> * The comparison of TGI✗/SFI✗/DFP✗ and TGI✓/SFI✗/DFP✗ demonstrates that TGI brings improvements to the overall performance, raising the average accuracy from 62.2% to 77.9%, even when used alone. This shows the crucial role of TGI in our framework.
> * The comparison of TGI✗/SFI✓/DFP✗ and TGI✓/SFI✓/DFP✗ demonstrates that adding TGI on top of SFI further improves the performance, boosting the average accuracy from 74.4% to 82.2%.
> * The comparison of TGI✗/SFI✗/DFP✓ and TGI✓/SFI✗/DFP✓ demonstrates that TGI can also further enhance when combined with DFP, enhancing the average accuracy from 76.8% to 79.8%.
> * The comparison of TGI✗/SFI✓/DFP✓ and TGI✓/SFI✓/DFP✓ demonstrates that introducing TGI on the basis of SFI and DFP further enhances the performance from 82.5% to 91.2%.
>
> Similarly, for Semantic Feature Injection and Domain-specific Feature Preservation, the results can also demonstrate the effectiveness and necessity of each component in our framework. We will include a comprehensive analysis in the revised version.

---

> > ### Comment · Reviewer_mAks · 2025-08-09
> >
> > I sincerely appreciate the authors for providing the additional experimental results. Most of my concerns have been addressed.

---

### Official Review · Reviewer_LADn · 2025-06-27

**Clarity:** 3
**Significance:** 1
**Originality:** 2
**Rating:** 4
**Confidence:** 5

**Summary:**

This paper presents a method called Diffusion-Driven Progressive Target Manipulation (**DPTM**) to tackle the challenge of source-free domain adaptation (SFDA). Specifically, DPTM partitions the target data into trusted and non-trusted sets based on the uncertainty of pseudo-label predictions, quantified by the entropy of the model outputs. A progressive training strategy is then introduced to iteratively refine the model and enhance adaptation performance.

**Questions:**

1. While it is common to use Fast Fourier Transform (FFT) to extract domain-specific features, I am also curious about the effectiveness of using an image encoder for this purpose. It would be helpful if the authors could provide some insight into how the results compare between these two approaches.

2. I am wondering a more detailed comparison between this work and generation-based SFDA methods. Specifically, generating features and generating samples are fundamentally different, particularly in terms of computational cost. As far as I am aware, the generation-based methods listed in this paper do not generate samples.

3. It is unclear why similar colors are used for target data and target-guided initialization, as well as for target model and semantic feature injection in Figure 1. This similarity may cause confusion and reduce the overall clarity of the figure. Additionally, it could benefit from a clearer and more informative layout. For example, explicitly highlighting the interaction relationships among different color blocks would enhance the reader's understanding.

**Ethical Concerns:**

["NO or VERY MINOR ethics concerns only"]

**Final Justification:**

The authors have provided clarifications and experiments that address most of my concerns, so I raised my score to 4.

**Limitations:**

The major limitation is similar to the first weakness mentioned above.

**Paper Formatting Concerns:**

I have not notice any formatting issues in this paper.

**Quality:**

2

**Strengths And Weaknesses:**

Strenths:

1. **Good Performance**. The proposed DPTM achieves state-of-the-art performance across several widely used SFDA benchmarks, including Office-31, Office-Home, VisDA, and DomainNet-126. In particular, DPTM achieves an avearge classification accuracy of 91.2% on Office-Home, largely surpassing previous approches.

2. **Clear Presentation**. The presentation of the ideas, framework, and experimental results is clear, making the paper easy to read and follow.

Weaknesses:

1. **Limited Contribution to the Community**.
One of my main concerns is that in the current era of large multimodal language models (MLLMs), the challenge of domain adaptation has been significantly mitigated by their powerful generalization capabilities. While I acknowledge that domain adaptation still holds practical value in specific application areas with insufficient data, such as the medical domain. Furthermore, the proposed method appears to be a straightforward combination of existing techniques, such as leveraging uncertainty from model outputs, employing a strong image generation model, applying FFT, and iteratvie refinement in , to improve performance. While I appreciate the impressive performance demonstrated on the Office-Home dataset, I believe the paper lacks sufficient novelty and theoretical contribution to meet the standards for acceptance at a top-tier conference like NeurIPS.

2. **Similarity to Existing Work**.
Using a powerful diffusion model to generate pseudo-labeled samples and incorporating them as external data for training has already been explored, such as [1] and [2].

3. **Lack of Visualization Results**.
It is recommended to include some visualization results illustrating the domain discrepancy. These visualizations would help support the claim that the proposed method effectively reduces domain discrepancy, as performance improvement alone provides only a limited perspective.

[1] Chopra, Shivang, et al. "Source-free domain adaptation with diffusion-guided source data generation." arXiv preprint arXiv:2402.04929 (2024).

[2] Peng, Duo, et al. "Unsupervised domain adaptation via domain-adaptive diffusion." IEEE Transactions on Image Processing (2024).

---

> ### Author Rebuttal · Authors · 2025-07-31
>
> ### Q1: Limited contribution to the community.
> First of all, we have to emphasize that we provide in this paper a novel paradigm that leverages diffusion-based generation to directly approximate the target domain itself. This paradigm significantly differs from existing methods that rely heavily on pseudo-labeling or attempt to generate pseudo-source data and overcomes the intrinsic domain gap by nature. Experimental results also demonstrate incredibly superior performance as strong support of our paradigm.
>
> In the field of domain adaptation, SFDA usually suffers from degraded performance caused by the lack of source data, which has not yet been addressed by existing methods. In this paper, we provide the key insight of leveraging diffusion-based generation to directly approximate the target domain rather than being restricted by making up the missing source data. We then realize this insight in a robust and generalizable manner and achieve strong empirical performance across diverse datasets. More importantly, this insight itself may open up new research directions for DA beyond SFDA, including broader scenarios such as UDA or even settings where the source model is entirely inaccessible. We have also made a detailed explanation on the practicality of our method in Response to Q1 by Reviewer mAks.
>
> ### Q2: Similarity to Existing Work.
> In SFDA, generating extra samples is a popular research paradigm due to the inherent data scarcity. Nevertheless, the fact that our method follows this paradigm should not compromise its novelty. The two works cited by the reviewer are **fundamentally different from our method**.
>
> * DM-SFDA: This method follows a pseudo-source data generation paradigm. We have already discussed the key differences between our approach and pseudo-source-based SFDA methods in Section 1. In terms of implementation, DM-SFDA relies heavily on complex fine-tuning of a diffusion model, which is totally different from our mechanism. Moreover, its performance is significantly lower than that of our method, as reported in Response to Q3 by Reviewer fShy.
>
> * DAD: **This work addresses UDA with source data, rather than SFDA without source data.** In fact, it focuses on leveraging source domain data effectively, while our method operates under a source-free setting where the source data is completely inaccessible, and we do not attempt to generate pseudo-source data.
>
> To the best of our knowledge, there are no existing SFDA methods that adopt the same design or similar specific mechanisms as ours.
>
>
> ### Q3: Lack of Visualization Results.
> Thanks for the suggestion, we first conduct Grad-CAM visualization using our SFDA model, and compare the results with those obtained by the source model. The results demonstrate that the attention of our model focuses on the target object, instead of other unrelated regions. Besides, we conduct t-SNE analysis on target domain feature distribution, and our method exhibits the least category aliasing.
>
>
> ### Q4: Using an image encoder to replace FFT.
> We use FFT to obtain a pseudo-image in the Target-guided Initialization stage and a desirable latent in both Semantic Feature Injection and Domain-specific Feature Preservation. In this case, using an image encoder introduces several challenges:
> * First, the extracted features often have altered dimensions, making it difficult to reconstruct the original pseudo-image or latent representation without an accompanying decoder.
> * Second, this necessitates pretraining both the image encoder and decoder to extract and reconstruct frequency components. However, it is hard to guarantee effective pretraining using the unlabeled target data, especially on small-scale datasets. This can significantly degrade the performance of our method, while adding extra complexity and computational costs.
>
> In contrast, our experiments demonstrate that FFT is simple yet effective. Even in real-world scenarios, it introduces negligible overhead and does not hinder the overall performance of the algorithm. We have evaluated the performance of our methods on real-world scenarios VLCS, using unsupervised model selection without using target labels, as elaborated in Response to Q1 by Reviewer mAks.
>
> ### Q5: Generation-based methods listed in this paper do not generate samples.
> We list 4 generation-based methods, including CPGA, ASOGE, ISFDA, and  PS. Actually, two of them, *i.e.*, ISFDA and PS, generate samples.
>
> ### Q6: Similar colors are used in Figure 1.
> We will correct the confusion in Figure 1.

---

> > ### Comment · Reviewer_LADn · 2025-08-05
> >
> > Thank you for the authors' efforts in providing clarification. The rebuttal has addressed most of my concerns. However, I am still curious about why the FFT is effective in the proposed method. Could the authors elaborate further on this aspect? Besides, I am wondering the motivation of the design of Target-guided Initilization in Figure 1. The authors argue that Target-guided Initialization is designed to obtain an effective sampling starting point. Becase such design can intialize a noise maintaining the semantics of target samples? Would it misalign the theory of a stadard denoising diffusion model?

---

> ### Author Response · Authors · 2025-08-07
>
> We sincerely thank the reviewer for the response. We will respond to these questions in detail below. To better address the concerns, we first provide a more in-depth analysis of the Target-guided Initialization,  and then provide a further explanation of the effectiveness of the FFT component.

---

> ### Author Response · Authors · 2025-08-07
> **New Q1: Target-guided Initilization (1/2)**
>
> ## New Q1: Target-guided Initilization
>
> ## Reviewer comment
> I am wondering the motivation of the design of Target-guided Initilization in Figure 1. The authors argue that Target-guided Initialization is designed to obtain an effective sampling starting point. Because such design can initialize a noise maintaining the semantics of target samples? Would it misalign the theory of a standard denoising diffusion model?
>
> ## Response
> ### Motivation of Target-guided Initialization
> We design Target-guided Initialization to **construct a semantically neutral yet target-domain-aware sampling starting point**. The motivation is based on recent findings about the importance of the sampling starting point and the signal-leak bias in diffusion models. We elaborate on this motivation in the following steps:
>
> 1. **Sampling starting point matters:**
>
> For diffusion models, prior research has shown that the sampling starting point can have a significant impact on the final generation results, and that carefully designing this starting point can lead to improved performance [1][2][3][4].
>
> 2. **Signal-leak bias in diffusion models:**
>
> Specifically, [1] investigates the relationship between **the generated image** and **the sampling starting point** of the diffusion model, and identifies a phenomenon termed **signal-leak bias**:
>
> * During the **training process**, the sampling starting point is obtained by corrupting an image with $T$ steps of Gaussian noise. Even when $T$ is large, the low-frequency components of the image are not entirely destroyed. As a result, the sampling starting point retains part of the original image's low-frequency information. During training, diffusion models learn to exploit this low-frequency signal for the sake of minimizing the training loss. Consequently, the generated images tend to preserve low-frequency information that is present in the sampling starting point.
> * During the **inference process**, the diffusion model continues to utilize the low-frequency components of the sampling starting point to guide the generation process, which is referred to as **signal-leak bias**.
>
> 3. **Leveraging signal-leak bias for SFDA:**
>
> We observe that the signal-leak bias can be effectively leveraged to benefit the SFDA task. In domain adaptation and generalization, low-frequency components of images are considered to partly contribute to domain-specific features [5][6]. For each non-trust sample, since it belongs to the target domain, its low-frequency components can be considered to partly carry certain target-domain-specific features. Therefore, our method extracts the low-frequency components from the non-trust sample and uses them to form the sampling starting point of the diffusion model, allowing the generated image to inherit part of the target domain features via signal-leak bias, which **facilitates adaptation**.
>
> 4. **Semantic neutrality via high-frequency noise:**
>
> Besides, it has been shown that in diffusion models, high-frequency components are closely related to semantic detail patterns[7], such as object contours and textures. Since the real class label of the non-trust sample is unknown, we randomly assign a class label and aim to transform the non-trust sample into the assigned class via the generation process. As this transformation involves **a semantic transform process**, we choose to extract high-frequency components from a semantically neutral random Gaussian noise, ensuring that no class-specific information is inadvertently introduced at the sampling starting point.
>
> 5. **Combining both components to construct the sampling starting point:**
>
> We combine the aforementioned extracted low-frequency and high-frequency components, and perform IFFT to produce a semantically neutral pseudo-image carrying target domain features. This pseudo-image is then encoded into a latent using the VAE encoder of the diffusion model. The obtained latent is denoted as $\hat{\mathbf{z}}_{0}$ in the manuscript.
>
> Finally, we perform **the DDPM forward process** on $\hat{\mathbf{z}}_{0}$ by adding $T$ steps of Gaussian noise, resulting in a noisy latent representation $\mathbf{z}_T$. This process corresponds to Equation (5) in the manuscript:
>
> $$ \mathbf{z}_T=\sqrt{\alpha_T} \hat{\mathbf{z}}_0+\sqrt{1-\alpha_T} \boldsymbol{\epsilon}, \quad \boldsymbol{\epsilon} \sim \mathcal{N}(\mathbf{0}, \mathbf{I}) .$$
>
> The resulting $\mathbf{z}_T$ is employed as **the sampling starting point** of the diffusion model. According to the aforementioned signal-leak bias, the DDPM forward process preserves low-frequency components while corrupting high-frequency components. This allows the sampling starting point $\mathbf{z}_T$ to **retain target domain features while becoming more semantically neutral**.

---

> > ### Author Response · Authors · 2025-08-07
> > **New Q1: Target-guided Initilization (2/2)**
> >
> > ## Alignment with the standard denoising diffusion model
> >
> > According to the aforementioned analysis, we first combine the extracted high-frequency and low-frequency components via IFFT to construct a pseudo-image. This pseudo-image is then encoded into a latent $\hat{\mathbf{z}}_0$ using a VAE. Finally, we perform **DDPM forward to obtain the sampling starting point** $\mathbf{z}_T$ by:
> >
> > $$ \mathbf{z}_T=\sqrt{\alpha_T} \hat{\mathbf{z}}_0+\sqrt{1-\alpha_T} \boldsymbol{\epsilon}, \quad \boldsymbol{\epsilon} \sim \mathcal{N}(\mathbf{0}, \mathbf{I}) .$$
> >
> > Therefore, our sampling starting point $\mathbf{z}_T$ is obtained via the DDPM forward process, which adds Gaussian noise to a clean latent $\hat{\mathbf{z}}_0$ according to the theory of a standard denoising diffusion model. **Due to the property of the DDPM forward process**, $\mathbf{z}_T$ still follows $\mathcal{N}(\mathbf{0}, \mathbf{I})$, which is a direct consequence of **the reparameterization trick**. Therefore, our sampling starting point **aligns with the theoretical assumptions of a standard denoising diffusion model**.

---

> > > ### Author Response · Authors · 2025-08-07
> > > **New Q2: FFT**
> > >
> > > ## New Q2: FFT
> > >
> > > ## Reviewer comment
> > > I am still curious about why the FFT is effective in the proposed method. Could the authors elaborate further on this aspect?
> > >
> > > ## Response
> > >
> > > Frequency-domain-based decomposition has been shown to be an effective strategy for forming sampling starting points in diffusion models. Prior research has demonstrated the effectiveness of such frequency-domain-based decomposition [1][3][4]. As analysed above, the FFT in our method allows us to independently manipulate low-frequency and high-frequency signals at a fine-grained level.
> > >
> > > To better validate the effectiveness of our design, we provide more component-wise additional ablation studies as shown below. For clarity in the tables, we refer to Target-guided Initialization, Semantic Feature Injection, and Domain-specific Feature Preservation as TGI, SFI, and DFP, respectively.
> > >
> > >
> > > | TGI | SFI | DFP | Ar→Cl | Ar→Pr | Ar→Rw | Cl→Ar | Cl→Pr | Cl→Rw | Pr→Ar | Pr→Cl | Pr→Rw | Rw→Ar | Rw→Cl | Rw→Pr | Avg. |
> > > |-----|-----|-----|-------|-------|-------|-------|-------|-------|-------|-------|-------|-------|-------|-------|------|
> > > | ✗   | ✗   | ✗   | 50.1  | 67.9  | 74.4  | 55.2  | 65.2  | 67.2  | 53.4  | 44.5  | 74.1  | 64.2  | 51.5  | 78.7  | 62.2 |
> > > | ✓   | ✗   | ✗   | 69.2  | 86.2  | 82.2  | 74.6  | 87.8  | 80.7  | 76.7  | 67.8  | 82.4  | 73.5  | 66.9  | 87.1  | 77.9 |
> > > | ✗   | ✓   | ✗   | 59.6  | 80.8  | 82.3  | 67.9  | 83.1  | 80.2  | 64.6  | 68.1  | 81.7  | 70.6  | 67.9  | 86.5  | 74.4 |
> > > | ✗   | ✗   | ✓   | 67.8  | 87.6  | 82.7  | 68.7  | 84.0  | 80.6  | 70.7  | 67.6  | 82.4  | 74.1  | 67.1  | 88.7  | 76.8 |
> > > | ✗   | ✓   | ✓   | 75.9  | 89.4  | 88.3  | 78.9  | 87.9  | 87.0  | 76.0  | 74.1  | 87.8  | 80.0  | 75.9  | 89.1  | 82.5 |
> > > | ✓   | ✗   | ✓   | 72.9  | 88.0  | 84.1  | 77.5  | 88.3  | 83.6  | 76.8  | 69.0  | 84.0  | 75.7  | 68.8  | 88.4  | 79.8 |
> > > | ✓   | ✓   | ✗   | 70.2  | 89.9  | 85.5  | 81.1  | 89.4  | 89.3  | 80.7  | 70.7  | 86.3  | 80.3  | 72.1  | 91.0  | 82.2 |
> > > | ✓   | ✓   | ✓   | 86.7  | 94.2  | 92.8  | 91.5  | 94.0  | 92.6  | 90.6  | 86.4  | 92.8  | 90.5  | 87.1  | 94.7  | 91.2 |
> > >
> > > As is shown in the results, for example, Target-guided Initialization, which employs FFT, is a significant component of our method. We provide a detailed analysis below, which demonstrates the effectiveness of FFT.
> > > * The comparison of TGI✗/SFI✗/DFP✗ and TGI✓/SFI✗/DFP✗ demonstrates that TGI brings improvements to the overall performance, raising the average accuracy from 62.2% to 77.9%, even when used alone. This shows the crucial role of TGI in our framework.
> > > * The comparison of TGI✗/SFI✓/DFP✗ and TGI✓/SFI✓/DFP✗ demonstrates that adding TGI on top of SFI further improves the performance, boosting the average accuracy from 74.4% to 82.2%.
> > > * The comparison of TGI✗/SFI✗/DFP✓ and TGI✓/SFI✗/DFP✓ demonstrates that TGI can also further enhance when combined with DFP, enhancing the average accuracy from 76.8% to 79.8%.
> > > * The comparison of TGI✗/SFI✓/DFP✓ and TGI✓/SFI✓/DFP✓ demonstrates that introducing TGI on the basis of SFI and DFP further enhances the performance from 82.5% to 91.2%.
> > >
> > >
> > > [1]Everaert M N et al. Exploiting the signal-leak bias in diffusion models. WACV 2024.
> > >
> > > [2]Xu R et al. StyleSSP: Sampling StartPoint Enhancement for Training-free Diffusion-based Method for Style Transfer. CVPR 2025.
> > >
> > > [3]Koo G et al. Flexiedit: Frequency-aware latent refinement for enhanced non-rigid editing. ECCV 2024.
> > >
> > > [4]Wu T et al. Freeinit: Bridging initialization gap in video diffusion models. ECCV 2024.
> > >
> > > [5]Guo J et al. Aloft: A lightweight mlp-like architecture with dynamic low-frequency transform for domain generalization. CVPR 2023.
> > >
> > > [6]Wang J et al. Domain generalization via frequency-domain-based feature disentanglement and interaction. ACM MM 2022.
> > >
> > > [7]Qian Y et al. Boosting diffusion models with moving average sampling in frequency domain. CVPR 2024.

---

> > > > ### Comment · Reviewer_LADn · 2025-08-08
> > > >
> > > > Thank you for authors' detailed response. I have no questions.

---

### Official Review · Reviewer_pUPg · 2025-06-29

**Clarity:** 4
**Significance:** 3
**Originality:** 3
**Rating:** 4
**Confidence:** 5

**Summary:**

-	In the source free domain adaptation (SFDA) task that uses only unlabeled target data and a source model pretrained with labeled source data, the authors propose a generation-based framework named Diffusion-Driven Progressive Target Manipulation (DPTM) to effectively utilize the non-trust set with high uncertainty among the target model's prediction results in the target domain by supplementing it with a diffusion model.

**Questions:**

-	When generating samples of the target domain and applying them to train the target SFDA model, it seems that the size of the target SFDA model will have a great impact on the performance. Is there any analysis related to this?
-	A description and classification of the types of benchmark methods is needed.
-	The performance improvement is quite extraordinary, and theoretical analysis is essential. The proposed method should quantitatively measure how well the samples generated by the diffusion model represent the target domain.
-	How specifically was the grid search performed to find threshold 𝐸?
-	At line 255, it says ‘we do not specifically optimize the search space’. What does this mean?

**Ethical Concerns:**

["NO or VERY MINOR ethics concerns only"]

**Final Justification:**

Through the rebuttal process, we have confirmed all of the following.
1. The computational complexity of the proposed method
2. Performance variation depending on target model size
3. The reliability of the proposed method's entropy-based Trust and Non-trust Partition


Based on this, we have decided to maintain our original decision (4: Borderline accept).

**Limitations:**

-	The limitations of the proposed technique are stated in the Appendix.

**Paper Formatting Concerns:**

- The font of characters such as (a) used in the figure needs to be unified.
- Some errors in the reference entry need to be corrected.

**Quality:**

3

**Strengths And Weaknesses:**

Strengths:

-	The proposed method of combining the diffusion model with SFDA significantly improves the performance of SFDA on existing datasets.
-	It is an ingenious approach to use the existing widely used DA technique based on the idea that low-frequency of an image represents domain information and high-frequency represents semantic information, to the starting point data of the diffusion model.

Weaknesses

-	The proposed method aims at SFDA, but it seems unfair to compare the performance with existing SFDA methods because it additionally uses a diffusion model with high computational cost in addition to the source model.
-	The computational complexity of the proposed method is expected to be considerably high, and a detailed analysis is required, including comparison with benchmark methods.
-	The reliability of the proposed method's entropy-based Trust and Non-trust Partition is not high. Therefore, the evaluation of the reliability of the entropy-based Trust and Non-trust Partition is required. In fact, the pseudo labeling accuracy of the two sets of needs to be evaluated, and the SFDA performance of the proposed method needs to be evaluated depending on the partition accuracy.
-	The performance improvement is unrealistically high, so it is necessary to present and compare the performance when the model is trained with all samples from the target domain. In addition, considering the current performance improvement of the proposed method, it is expected that the performance of the proposed method should be higher when all samples from the target domain are set as non-trust set.
-	The supplementary material contains repeated experimental results from the manuscript and does not contain the necessary new experimental results mentioned above. In particular, there should be a wider range of examples of samples generated via the Diffusion model in the target domain

---

> ### Author Rebuttal · Authors · 2025-07-31
>
> ### Q1: Unfair comparison
>
> * A prevalent paradigm involves employing generative models to synthesize data, as target domain data scarcity is a fundamental challenge in SFDA settings. Consistent with other generation-based SFDA methods, our method strictly adheres to SFDA settings by completely avoiding any use of source domain data during the generation process.
> * The use of external models for adaptation is common in SFDA, as demonstrated by established methods like DIFO and ProDe in Section 4.1.
> * We only utilize the diffusion model for adaptation training. In the inference stage, we do not use the diffusion model.
>
> ### Q2: Cost analysis
>
> Please refer to the Response to Q3 by Reviewer Rmwo for detailed cost analysis.
>
> ### Q3: Reliability of entropy-based trust and non-trust partition
> Entropy-based selection of reliable pseudo-labels of target samples is commonly used in SFDA, and its effectiveness is demonstrated in prior studies [1]. To further evaluate its reliability, we report the trust set accuracy evolving with $r$ on the Office-Home dataset as below ($r$ denotes the $r$-th refinement iteration, where $r=1,2,...R$, and in our experiments we set $R=10$). Figure 2(a) shows that the size of the trust set grows as $r$ increases. Remarkably, the trust accuracy remains consistently high across all tasks using a total of 10 refinement iterations. Besides, when $r$ is small, the trust set accuracy may not be high, but will increase to a high value as $r$ grows. For example, in tasks like Pr→Ar and Pr→Cl, trust accuracy is below 90% when $r=2$ but reaches higher than 97% when $r=10$. These results validate the effectiveness of our method that could correct errors in the trust set with the growth of $r$. Note that the non-trust set accuracy does not affect the performance of our method, as we completely discard the original pseudo-labels of non-trust samples.
>
>
> | $r$ | Ar→Cl | Ar→Pr | Ar→Rw | Cl→Ar | Cl→Pr | Cl→Rw | Pr→Ar | Pr→Cl | Pr→Rw | Rw→Ar | Rw→Cl | Rw→Pr |
> |-----|:-----:|:-----:|:-----:|:-----:|:-----:|:-----:|:-----:|:-----:|:-----:|:-----:|:-----:|:-----:|
> | $1$  | 95.4  | 97.6  | 98.5  | 99.1  | 100.0 | 99.8  | 100.0 | 100.0 | 99.8  | 100.0 | 97.4  | 99.1  |
> | $2$  | 85.3  | 95.0  | 97.2  | 88.8  | 93.9  | 96.1  | 85.9  | 83.8  | 96.6  | 87.5  | 86.4  | 96.3  |
> | $3$  | 94.5  | 98.3  | 97.6  | 91.2  | 98.2  | 97.3  | 90.0  | 91.9  | 97.8  | 92.9  | 93.1  | 98.2  |
> | $4$  | 94.2  | 98.0  | 98.1  | 94.3  | 98.1  | 97.8  | 93.6  | 95.2  | 98.0  | 94.4  | 95.7  | 98.4  |
> | $5$  | 96.6  | 99.0  | 98.7  | 95.1  | 99.0  | 98.5  | 94.4  | 95.8  | 98.2  | 95.6  | 95.7  |  99   |
> | $6$  | 96.3  | 99.1  | 98.6  | 95.7  | 99.1  | 98.6  | 96.1  | 96.8  | 98.6  | 95.0  | 97.3  |  99   |
> | $7$  | 97.1  | 99.4  | 98.5  | 96.9  | 99.4  | 98.5  | 97.1  | 96.8  | 98.7  | 96.8  | 96.6  | 99.2  |
> | $8$  | 97.2  | 99.3  | 99.2  | 97.0  | 99.2  | 99.2  | 96.6  | 97.9  | 98.8  | 95.7  | 97.2  | 99.5  |
> | $9$  | 98.6  | 99.5  | 99.2  | 97.0  | 99.6  | 99.2  | 97.0  | 98.1  | 98.9  | 97.3  | 97.8  | 99.3  |
> | $10$ | 97.5  | 99.4  | 98.7  | 97.5  | 99.6  | 99.1  | 97.2  | 97.7  | 99.4  | 97.3  | 98.5  | 99.5  |
>
> [1]Liang J et al. Source data-absent unsupervised domain adaptation through hypothesis transfer and labeling transfer. TPAMI 2021.
>
> ### Q4: Set all target samples to non-trust set.
> The high performance gain of our method stems from two key mechanisms: i) progressively expanding the trust set with high-accuracy pseudo-labels to allow the SFDA model to learn real target domain features, and ii) progressively reducing the manipulated non-trust set. Mechanism ii) is critical to prevent features from the synthetic domain becoming dominant, since there is an inherent gap that persists between the synthetic and real target domains (even though we employ alignments to bridge the gap). This phenomenon fundamentally stems from the inherent domain gap between synthetic and real data, a well-documented challenge that has been rigorously demonstrated in prior work [2].
>
> Therefore, canceling the trust set could cause degraded performance, since we could only obtain features from the synthetic domain. We validate it with an empirical study on the first 4 Office-Home tasks (Ar→Cl, Ar→Pr, Ar→Rw, and Cl→Ar), where we cancel the trust set and set all target samples to the non-trust set.  $r$ denotes the $r$-th refinement iteration, and we report results of $r=1,2,...,5$. As shown below, when using only non-trust samples, the performance is not improved as $r$ grows.
>
> |     | Ar$\to$Cl | Ar$\to$Pr | Ar$\to$Rw | Cl$\to$Ar |
> |-----|:---------:|:---------:|:---------:|:---------:|
> | source |    50.1   |    67.9   |    74.4   |    55.2   |
> | $r=1$ |    56.4   |    78.7   |    81.7   |    63.0   |
> | $r=2$ |    55.9   |    78.5   |    81.4   |    62.8   |
> | $r=3$ |    55.9   |    78.3   |    81.6   |    62.7   |
> | $r=4$ |    55.8   |    78.1   |    81.4   |    62.7   |
> | $r=5$ |    55.7   |    77.9   |    81.3   |    62.7   |
>
> [2] Akagic A et al. Exploring the Impact of Real and Synthetic Data in Image Classification: A Comprehensive Investigation Using CIFAKE Dataset. CoDIT 2024.
>
> ### Q5: Appendix.
> We will remove repeating experimental results and add more generated samples.
>
> ### Q6: The size of the target SFDA model.
> Table A.1 shows results on VisDA with ResNet-101. Below, we provide extra results with ResNet-50 to further demonstrate the superior performance of our method.
>
> | Method     | plane | bcycl |  bus |  car | horse | knife | mcycl | person | plant | sktbrd | train | truck | Perclass |
> |------------|:-----:|:-----:|:----:|:----:|:-----:|:-----:|:-----:|:------:|:-----:|:------:|:-----:|:-----:|:--------:|
> | Source-R50 |  79.7 |  35.5 | 44.5 | 63.8 |  62.0 |  25.0 |  86.9 |  26.6  |  77.6 |  30.0  |  94.7 |  12.7 |   53.3   |
> | Ours-R50   |  99.5 |  96.1 | 93.5 | 82.5 |  98.3 |  99.2 |  96.3 |  96.8  |  98.8 |  98.5  |  98.1 |   81  |   94.9   |
>
> ### Q7: Description and classification of the types of benchmark methods.
> We will further classify and reorganize benchmark methods.
>
> ### Q8:  Quantitatively measure how well the samples generated by the diffusion model represent the target domain.
>
> We conduct cross-domain validation by training models on real target domain data and evaluating them on i) our generated target domain (1st row) and ii) Other Office-Home domains for reference (2nd row). For example, the model trained on real Art domain data (first column) achieves 84.2% accuracy on our synthetic Art samples, and 64.1% average accuracy on Clipart, Product, and RealWorld domains. The results indicate that our generated samples closely approximate the real target domain, and the domain gap between synthetic and real data is significantly smaller than the natural gap between Office-Home domains.
>
> |     | Ar | Cl | Pr | Rw |
> |-----|:---------:|:---------:|:---------:|:---------:|
> | real target->synthetic target             |   84.2	|81.4	    |87.9	    |89.4   |
> | real target->other Office-Home domain     |   64.1    |	62.5|	57.3|	64.8  |
>
>
> ### Q9: How specifically was the grid search performed to find threshold $E$?
> We use fixed values that $E=0.01$ and $R=10$ for all the results in Table 1-3 without any change. We only perform grid search once on Office-Home to explore $E\in[0.01, 0.007, 0.005, 0.001]$ and $R\in[1,...,10]$ and then fix the values $E=0.01$ and $R=10$ for all the datasets. We offer a detailed explanation in the Response to Q1 by Reviewer mAks, and will clarify it in the final version.
>
> ### Q10: What does ‘we do not specifically optimize the search space’ mean?
> We intend to mean we do not exhaustively search all the datasets for optimal values of $E$ and $R$, as specified in response to Q9. In fact, using larger $E$ or $R$ (*e.g.*, $E>0.01$ or $R>10$) could obtain better results.

---

> ### Comment · Reviewer_pUPg · 2025-08-01
>
> The responses were generally insincere and did not fully address my questions. Below are some details.
>
> Q1: Unfair comparison:
> The existing methods mentioned by the authors did not use genertive models like diffusion model for generating samples in the target domain. The proposed method utilizes a diffusion model capable of generating the samples in the target domain. Please tell us about the fairness of the comparison from this perspective.
>
> Q2: Cost analysis:
> There’s no answer for my question in Response to Q3 by Reviewer Rmwo. Maybe response to Q4 is correct. This response simply mentions the computational complexity of the proposed method, and the analysis is very low quality and insincere. A table should be provided comparing the training time, inference time, and memory [GB] usage compared to benchmark methods.
>
> Where’s the response about the performance comparison with the model that is trained with all samples from the target domain?
> In response to Q6, I don't understand why the authors gave me this answer. What does the Method in each row mean? Where’s Table A? Please review the first question again.

---

> > ### Author Response · Authors · 2025-08-03
> >
> > We apologize for the lack of clarity in our initial response. We understand that several of our earlier answers did not adequately address your questions, and we deeply regret that this may have come across as insincere.
> >
> > During the rebuttal period, we took your comments seriously. However, due to time constraints and a lack of full understanding of some of the concerns at the time, we failed to respond appropriately. We have since realized that important clarifications and analyses were missing from our earlier reply, and we have now conducted additional analyses to directly address the issues you raised.
> >
> > Your feedback was invaluable in helping us identify and correct these shortcomings. Below, we provide detailed, point-by-point responses with expanded explanations, concrete comparisons, and new analyses addressing your concerns.

---

> > ### Author Response · Authors · 2025-08-03
> > **New Q1: Unfair comparison (1/2)**
> >
> > ## New Q1: Unfair comparison
> >
> > ## Reviewer comment
> >
> > The existing methods mentioned by the authors did not use genertive models like diffusion model for generating samples in the target domain. The proposed method utilizes a diffusion model capable of generating the samples in the target domain. Please tell us about the fairness of the comparison from this perspective.
> >
> > ## Response
> >
> > We appreciate your valid concern regarding comparison fairness. We would like to clarify this issue from three perspectives:
> >
> > 1. No prior works in SFDA use generative models to generate target samples.
> >
> >  In this paper, we propose a novel paradigm that leverages a generative model (specifically, we use the diffusion model) to generate target samples to address SFDA, which directly approximates the target domain itself. To the best of our knowledge, this is the first work in SFDA that employs this paradigm, and there are no prior works in SFDA that adopt generative models for generating target samples. This paradigm is also one of our key contributions.
> >
> > 2. Performing generation does not violate the SFDA setting.
> >
> >  According to the standard SFDA setting, source data is inaccessible during adaptation, but it is free to use the source model and unlabeled target data. Performing generation in the adaptation phase is allowed, as long as the source data is not accessed.
> >
> > Several existing SFDA methods also incorporate generation. Among these, CPGA [1] and ASOGE [2] generate features, while ISFDA [3] and PS [4] generate samples. As our method also generates samples, we further explain our difference with ISFDA [3] and PS [4]. ISFDA [3] employs augmentation mechanisms (e.g., rotation, invert, random clip) to generate harder samples for the SFDA model. PS [4] also employs data augmentation but generates pseudo-source samples. Thus, none of them use diffusion models or generate target samples.
> >
> > To further enhance the fairness of our comparison, we additionally considered a recent arXiv paper, DM-SFDA [5]. DM-SFDA [5] also uses a diffusion model, but for generating pseudo-source data. Although the DM-SFDA method is not peer-reviewed, from the perspective of using the diffusion model, we included this method in comparisons as below for completeness. We present the comparison results between DM-SFDA [5] and our method on Office-31, Office-Home, and VisDA datasets, respectively. However, regarding the DomainNet-126 dataset, we are unable to obtain DM-SFDA's performance since DM-SFDA neither reports these results in its paper nor has been open-sourced. The results below demonstrate that our method also outperforms the diffusion-based method DM-SFDA.
> >
> > | Method  | Office-31 |      |      |      |      |       |      |
> > |---------|:---------:|:----:|:----:|:----:|:----:|:-----:|:----:|
> > |         |    A→D    |  A→W |  D→A |  D→W |  W→A |  W→D  | Avg. |
> > | DM-SFDA [5] |    97.7   | 99.0 | 82.7 | 99.3 | 83.5 | 100.0 | 93.7 |
> > | **Ours (DPTM)**   |    97.2   | 95.3 | 92.0 | 98.7 | 91.7 | 100.0 | 95.8 |
> >
> > | Method  | Office-Home |       |      |      |       |       |       |        |       |        |       |       |          |
> > |---------|:-----:|:-----:|:----:|:----:|:-----:|:-----:|:-----:|:------:|:-----:|:------:|:-----:|:-----:|:--------:|
> > |         |Ar→Cl | Ar→Pr | Ar→Rw | Cl→Ar | Cl→Pr | Cl→Rw | Pr→Ar | Pr→Cl | Pr→Rw | Rw→Ar | Rw→Cl | Rw→Pr | Avg.  |
> > | DM-SFDA [5]                                     |    68.5  |    89.6   |    83.3   |    70.0   |    85.8   |   87.4   |   71.3   |    69.6   |    88.2   |    77.8   |    68.5   |    88.7   |  79.5 |
> > | **Ours (DPTM)**                                    |    86.7   |    94.2   |    92.8   |    91.5   |    94.0   |    92.6   |    90.6   |    86.4   |    92.8   |    90.5   |    87.1   |    94.7   |  91.2 |
> >
> > | Method  | VisDA |       |      |      |       |       |       |        |       |        |       |       |          |
> > |---------|:-----:|:-----:|:----:|:----:|:-----:|:-----:|:-----:|:------:|:-----:|:------:|:-----:|:-----:|:--------:|
> > |         | plane | bcycl |  bus |  car | horse | knife | mcycl | person | plant | sktbrd | train | truck | Perclass |
> > | DM-SFDA [5] |  98.1 |  89.8 | 90.6 | 90.5 |  96.8 |  95.2 |  92.2 |  93.4  |  97.8 |  94.4  |  92.4 |  48.8 |   86.3   |
> > | **Ours (DPTM)**  |  99.5 |  97.1 | 96.2 | 93.0 |  99.2 |  99.2 |  98.8 |  97.7  |  99.3 |  99.7  |  98.6 |  93.1 |   97.6   |

---

> > > ### Author Response · Authors · 2025-08-03
> > > **New Q1: Unfair comparison (2/2)**
> > >
> > > 3. Using an external model does not violate the SFDA settings.
> > >
> > > The use of external models during adaptation is permitted in SFDA and has become a widely adopted paradigm in recent SFDA works. Apart from the above-mentioned generative models, several methods, including DIFO [6] and ProDe [7] (the current SOTA SFDA method), introduce a strong external perceptual model, CLIP, for SFDA. These methods are widely considered acceptable within SFDA settings, and their performance also remains lower than that of our method.
> > >
> > > Additionally, we would like to clarify that the diffusion model is used only during the adaptation training of the target model. Following comparative methods, we use ResNet-50 as the target model for Office-31, Office-Home, and DomainNet-126, and ResNet-101 for VisDA. After the adaptation training is completed, the diffusion model is no longer used, and only the target model is used for inference.
> > >
> > >
> > > We hope this revised explanation addresses your concerns about fairness. If any point remains unclear, we would be grateful for further clarification.
> > >
> > > [1]Qiu Z et al. Source-free domain adaptation via avatar prototype generation and adaptation, IJCAI 2021.
> > >
> > > [2]Cui C et al. Adversarial source generation for source-free domain adaptation. TCSVT 2023.
> > >
> > > [3]Mitsuzumi Y et al. Understanding and improving source-free domain adaptation from a theoretical perspective. CVPR 2024.
> > >
> > > [4]Du Y, Yang H, Chen M, et al. Generation, augmentation, and alignment: A pseudo-source domain based method for source-free domain adaptation. Machine Learning 2024.
> > >
> > > [5]Chopra S et al. Source-free domain adaptation with diffusion-guided source data generation. arXiv:2402.04929.
> > >
> > > [6]Yang S et al. Attracting and dispersing: A simple approach for source-free domain adaptation. NeurIPS 2022.
> > >
> > > [7]Tang S et al. Proxy denoising for source-free domain adaptation. ICLR 2025.

---

> > > > ### Author Response · Authors · 2025-08-03
> > > > **New Q2: Cost analysis (1/3)**
> > > >
> > > > ## New Q2: Cost analysis
> > > >
> > > > ## Reviewer comment
> > > >
> > > > There’s no answer for my question in Response to Q3 by Reviewer Rmwo. Maybe response to Q4 is correct. This response simply mentions the computational complexity of the proposed method, and the analysis is very low quality and insincere. A table should be provided comparing the training time, inference time, and memory [GB] usage compared to benchmark methods.
> > > >
> > > > ## Response
> > > >
> > > > We apologize for not providing a detailed cost analysis in tabular form in our initial response. This was primarily due to the fact that most of the baseline methods we compared against did not report their running time or memory usage in their original papers. Nevertheless, we fully understand your concern and agree that providing such a comparison is important for evaluating the practicality of the proposed method.
> > > >
> > > > To address this, we have compiled comparative experiments on cost analysis, including **training time** and **peak GPU memory usage** for our method and **other benchmark methods whose official code is open-sourced and runnable**.
> > > >
> > > > We first describe the settings of our cost analysis experiments:
> > > >
> > > > * **Datasets.** We conduct experiments on the Office-Home dataset (moderate scale) and the DomainNet-126 dataset (large scale), as both contain a sufficient number of samples to reasonably evaluate the computational efficiency of different methods.
> > > > * **Benchmarked methods.** We successfully ran the following benchmark methods: **SHOT, NRC, GKD, AdaCon, CoWA, SCLM, PLUE, TPDS, DIFO, and ProDe**. All methods were executed strictly following the instructions provided in their official code repositories.
> > > > * **Measurement protocol.** Due to time constraints, we ran each comparative method for one epoch per SFDA task, recorded the training time and peak GPU memory usage during that epoch, and multiplied the time by the number of total epochs provided in their official code repositories to estimate the full training time. As for **our own method**, we had detailed logs from prior experiments, and we report the actual training time and GPU memory usage based on our full training runs. And we ran all the tasks with a single NVIDIA Tesla V100.

---

> > > > > ### Author Response · Authors · 2025-08-03
> > > > > **New Q2: Cost analysis (2/3)**
> > > > >
> > > > > Training time (hours) in the Office-Home dataset is listed below. Office-Home is a medium-scale dataset that contains 4 domains, including Art (Ar, 2427 images), Clipart (Cl, 4365 images) , Product (Pr, 4439 images), and Real-World (Rw, 4357 images).
> > > > >
> > > > > | Method | Ar→Cl | Ar→Pr | Ar→Rw | Cl→Ar | Cl→Pr | Cl→Rw | Pr→Ar | Pr→Cl | Pr→Rw | Rw→Ar | Rw→Cl | Rw→Pr | Avg. |
> > > > > |--------|:-----:|:-----:|:-----:|:-----:|:-----:|:-----:|:-----:|:-----:|:-----:|:-----:|:-----:|:-----:|:----:|
> > > > > | SHOT   |  3.0  |  2.8  |  8.2  |  1.8  |  3.0  |  8.3  |  1.5  |  2.7  |  7.8  |  1.4  |  2.9  |  3.0  |  3.9 |
> > > > > | NRC    |  0.9  |  1.4  |  2.4  |  0.9  |  1.2  |  3.9  |  1.3  |  1.7  |  2.6  |  1.0  |  1.2  |  1.4  |  1.7 |
> > > > > | GKD    |  2.7  |  2.9  |  8.3  |  1.6  |  2.8  |  8.4  |  1.6  |  3.0  |  7.6  |  1.6  |  2.9  |  2.6  |  3.8 |
> > > > > | AdaCon |  0.2  |  0.2  |  0.5  |  0.1  |  0.2  |  0.4  |  0.2  |  0.2  |  0.4  |  0.1  |  0.2  |  0.2  |  0.2 |
> > > > > | CoWA   |  0.9  |  2.4  |  3.5  |  0.6  |  1.2  |  2.6  |  0.8  |  1.6  |  1.5  |  1.0  |  1.2  |  1.5  |  1.6 |
> > > > > | SCLM   |  2.9  |  3.2  |  7.9  |  1.5  |  3.0  |  9.2  |  1.6  |  2.8  |  8.6  |  1.5  |  2.8  |  2.8  |  4.0 |
> > > > > | PLUE   |  0.2  |  0.3  |  0.6  |  0.2  |  0.5  |  0.5  |  0.1  |  0.2  |  0.4  |  0.2  |  0.2  |  0.2  |  0.3 |
> > > > > | TPDS   |  12.3 |  2.7  |  8.0  |  1.6  |  2.9  |  8.4  |  1.5  |  3.0  |  7.8  |  1.4  |  6.0  |  7.5  |  5.3 |
> > > > > | DIFO   |  6.9  |  7.2  |  22.4 |  4.1  |  7.3  |  25.5 |  4.3  |  7.7  |  26.5 |  4.4  |  7.3  |  7.8  | 11.0 |
> > > > > | ProDe  |  2.6  |  2.5  |  3.6  |  1.6  |  2.8  |  3.2  |  1.4  |  2.9  |  3.8  |  1.5  |  2.1  |  2.8  |  2.6 |
> > > > > | **Ours (DPTM)**  |  11.1 |  9.7  |   16  |  8.7  |  9.9  |  16.2 |  8.6  |  11.3 |  16.2 |  8.6  |  11.1 |  9.7  | 11.4 |
> > > > >
> > > > > Similarly, training time (hours) in the DomainNet-126 dataset is listed below. DomainNet-126 is a large-scale dataset that contains 4 domains, including clipart (C, 18523 images), painting (P, 10212 images), real (R, 69622 images), and sketch (S, 24147 images).
> > > > >
> > > > > | Method |  C→P |   C→R  |  C→S  |  P→C |   P→R  |  P→S  |  R→C |  R→P |  R→S  |  S→C |  S→P |   S→R  |  Avg.  |
> > > > > |--------|:----:|:------:|:-----:|:----:|:------:|:-----:|:----:|:----:|:-----:|:----:|:----:|:------:|:------:|
> > > > > | SHOT   | 10.9 |  392.8 |  51.5 | 34.9 |  437.9 |  68.4 | 37.3 | 14.2 |  66.4 | 37.5 | 12.3 |  391.7 |  129.7 |
> > > > > | NRC    |  1.0 |   5.4  |  2.3  |  3.3 |   9.5  |  4.6  |  2.1 |  1.5 |  3.9  |  2.9 |  1.1 |   4.8  |   3.5  |
> > > > > | GKD    | 12.2 | 4530.7 |  59.7 | 35.3 | 4134.7 |  67.1 | 37.3 | 10.4 |  72.7 | 38.6 | 13.1 | 4019.8 | 1086.0 |
> > > > > | AdaCon |  0.5 |   2.3  |  0.9  |  0.7 |   2.8  |  0.9  |  0.7 |  0.4 |  0.9  |  0.7 |  0.4 |   2.8  |   1.2  |
> > > > > | CoWA   |  1.4 |   6.2  |  5.3  |  4.8 |   7.8  |  5.4  |  4.4 |  2.8 |   6   |  3.2 |  2.1 |   9.3  |   4.9  |
> > > > > | PLUE   |  0.5 |   2.8  |  1.0  |  0.8 |   2.8  |  4.6  |  0.8 |  0.5 |  1.0  |  0.8 |  0.5 |   2.8  |   1.6  |
> > > > > | TPDS   | 11.3 |  383.4 |  52.7 | 39.3 |  426.9 |  68.3 | 39.4 | 14.1 |  69.2 | 40.2 | 13.9 |  406.3 |  130.4 |
> > > > > | DIFO   | 26.0 |  918.4 | 116.0 | 79.9 |  960.7 | 146.5 | 79.0 | 29.1 | 116.9 | 67.2 | 29.0 |  935.4 |  292.0 |
> > > > > | ProDe  |  5.1 |  27.1  |  17.0 | 15.8 |  48.1  |  16.7 |  8.8 |  8.1 |  16.9 |  7.8 |  5.2 |  28.5  |  17.1  |
> > > > > | **Ours (DPTM)**   | 13.7 |  45.3  |  28.5 | 22.2 |  43.6  |  30.1 | 22.2 | 13.7 |  24.8 | 20.4 | 13.3 |  43.5  |  26.8  |
> > > > >
> > > > > The peak GPU memory usage is listed below.
> > > > >
> > > > > | Method | Office-Home | DomainNet-126 |
> > > > > |--------|:-----------:|:-------------:|
> > > > > | SHOT   |     7GB     |      7GB      |
> > > > > | NRC    |     5GB     |      7GB      |
> > > > > | GKD    |     7GB     |      8GB      |
> > > > > | AdaCon |     14GB    |      14GB     |
> > > > > | CoWA   |     7GB     |      8GB      |
> > > > > | SCLM   |     7GB     |       -       |
> > > > > | PLUE   |     13GB    |      14GB     |
> > > > > | TPDS   |     7GB     |      7GB      |
> > > > > | DIFO   |     7GB     |      11GB     |
> > > > > | ProDe  |     12GB    |      13GB     |
> > > > > | **Ours (DPTM)**  |     10GB    |      12GB     |
> > > > >
> > > > > As for **inference time**, following most of the existing benchmark methods, our method uses only the target model (e.g., ResNet-50) for inference on the target domain. No additional modules or auxiliary models are involved during inference, so the inference time of our method is effectively the same as that of other benchmark methods using the same backbone.

---

> > > > > > ### Author Response · Authors · 2025-08-03
> > > > > > **New Q2: Cost analysis (3/3)**
> > > > > >
> > > > > > Based on the above comparison, we observe that our peak GPU memory usage is not higher than benchmark methods. As for training time, we provide a detailed analysis below:
> > > > > >
> > > > > > * On the medium-scale Office-Home dataset, our DPTM requires more training time than the benchmark methods. However, this is justified by the significant performance gain it achieves. Specifically, DPTM obtains an average accuracy improvement of 10.1% over current SOTA SFDA methods across all Office-Home tasks. Notably, on challenging tasks such as Ar→Cl, Pr→Cl, and Rw→Cl—where existing methods typically struggle—our method surpasses ProDe by 22.7%, 21.0%, and 21.6%, respectively. Moreover, the training time remains practically acceptable and does not increase to a level that hinders real-world applicability.
> > > > > >
> > > > > > * On the large-scale DomainNet-126 dataset, our DPTM shows lower training time compared to some benchmark methods. For example, methods like DIFO require particularly long training time, which limits their usability in real-world applications. In contrast, DPTM maintains practical training efficiency while also achieving superior performance over current SOTA methods.
> > > > > >
> > > > > > Therefore, DPTM presents a favorable balance between computational cost and adaptation performance, making it a practical and effective choice. All aforementioned experiments of DPTM were conducted on a single NVIDIA Tesla V100 GPU. Given the emergence of significantly more powerful GPUs in recent years, the training efficiency of our method can be further improved in practice.

---

> > > > > > > ### Author Response · Authors · 2025-08-03
> > > > > > > **New Q3: Trained with all samples from the target domain (1/2)**
> > > > > > >
> > > > > > > ## New Q3: trained with all samples from the target domain
> > > > > > >
> > > > > > > ## Reviewer comment
> > > > > > >
> > > > > > > Where’s the response about the performance comparison with the model that is trained with all samples from the target domain?
> > > > > > >
> > > > > > >
> > > > > > > ## Response
> > > > > > >
> > > > > > > We apologize for the lack of clarity in mapping each response to its corresponding question. Specifically, the question regarding the performance comparison with a model trained using all samples from the target domain was addressed in our original Response to Q4: “Set all target samples to non-trust set.” To further clarify and enhance our response, we now provide a more detailed explanation below.
> > > > > > >
> > > > > > > To better contextualize our understanding, we first quote the reviewer’s original comment:
> > > > > > > *  "it is necessary to present and compare the performance when the model is trained with all samples from the target domain. In addition, considering the current performance improvement of the proposed method, it is expected that the performance of the proposed method should be higher when all samples from the target domain are set as non-trust set."
> > > > > > >
> > > > > > > In our method, during each refinement iteration, we partition all target domain samples, assigning **some to the trust set** and **the rest to the non-trust set**. And the generation process is **only applied to the samples in the non-trust set**.
> > > > > > >
> > > > > > > Based on our understanding of the reviewer’s comment, specifically the suggestion to “set all samples from the target domain as non-trust set”, we interpret this as removing the trust set entirely. That is, in each refinement iteration, all target samples are assigned to the non-trust sets, and the generative process is applied to the entire target domain. For clarity, considering **our proposed method is DPTM**, **we refer to this modified variant as DPTM-AN ("AN" means All target samples assigned to the Non-trust set)**.
> > > > > > >
> > > > > > >
> > > > > > > We evaluate DPTM-AN on the Office-Home dataset. Due to the time limit, we only experiment on the first 4 tasks of the Office-Home dataset (Ar→Cl, Ar→Pr, Ar→Rw, and Cl→Ar). We present the comparison results of our DPTM and its aforementioned variant DPTM-AN below.
> > > > > > >
> > > > > > > |             | Ar$\to$Cl | Ar$\to$Pr | Ar$\to$Rw | Cl$\to$Ar |
> > > > > > > |-------------|-----------|-----------|-----------|-----------|
> > > > > > > | DPTM-AN     |    55.9   |    77.9   |    81.3   |    62.9   |
> > > > > > > | **Ours (DPTM)** |    86.7   |    94.2   |    92.8   |    91.5   |
> > > > > > >
> > > > > > > We observe a significant performance drop when using DPTM-AN compared to our original DPTM method, indicating that assigning all target samples to the non-trust set **does not lead to effective adaptation**. To provide further insight into this result, we analyze the performance trajectory of DPTM-AN over the first 5 refinement iterations (which means $r$ grows from 1 to 5, where $r$ denotes the $r$-th refinement iteration; recall that our method runs for 10 refinement iterations in total). For reference, we also include the performance of the source model before adaptation. The detailed performance results of DPTM-AN are presented below, which were also included in our initial response.
> > > > > > >
> > > > > > > | DPTM-AN trajectory | Ar$\to$Cl | Ar$\to$Pr | Ar$\to$Rw | Cl$\to$Ar |
> > > > > > > | ------------ |:---------:|:---------:|:---------:|:---------:|
> > > > > > > | source model |   50.1    |   67.9    |   74.4    |   55.2    |
> > > > > > > | $r=1$        |   56.4    |   78.7    |   81.7    |   63.0    |
> > > > > > > | $r=2$        |   55.9    |   78.5    |   81.4    |   62.8    |
> > > > > > > | $r=3$        |   55.9    |   78.3    |   81.6    |   62.7    |
> > > > > > > | $r=4$        |   55.8    |   78.1    |   81.4    |   62.7    |
> > > > > > > | $r=5$        |   55.7    |   77.9    |   81.3    |   62.7    |
> > > > > > >
> > > > > > > For comparison, we further present the performance trajectory of our original DPTM method below.
> > > > > > >
> > > > > > > | DPTM trajectory | Ar$\to$Cl | Ar$\to$Pr | Ar$\to$Rw | Cl$\to$Ar |
> > > > > > > |-----------------|-----------|-----------|-----------|-----------|
> > > > > > > | source model    |    50.1   |    67.9   |    74.4   |    55.2   |
> > > > > > > | $r=1$           |   60.2    |   80.9    |   82.8    |   67.8    |
> > > > > > > | $r=2$           |   72.6    |   87.2    |   85.7    |   73.5    |
> > > > > > > | $r=3$           |   75.3    |   89.7    |   87.6    |   77.7    |
> > > > > > > | $r=4$           |   79.0    |   91.8    |   89.1    |   81.0    |
> > > > > > > | $r=5$           |   81.5    |   92.5    |   89.8    |   83.9    |
> > > > > > >
> > > > > > > It can be observed that when all target samples are assigned to the non-trust set, the DPTM-AN performance quickly saturates and stops improving as the refinement iteration $r$ increases. In contrast, under our original DPTM design, where target samples are divided into a trust set and a non-trust set in each refinement iteration, the performance continues to improve steadily with increasing $r$.

---

> > > > > > > > ### Author Response · Authors · 2025-08-03
> > > > > > > > **New Q3: Trained with all samples from the target domain (2/2)**
> > > > > > > >
> > > > > > > > These results are reasonable, and we provide a detailed analysis below.
> > > > > > > >
> > > > > > > > In each refinement iteration, we re-partition all target domain samples using the current SFDA model. At the beginning of running DPTM when $r$ is small, only a small portion of target samples with reliable pseudo-labels are assigned to the trust set, while the majority are placed into the non-trust set. As $r$ increases, the trust set grows progressively, and correspondingly, the number of samples in the non-trust set decreases progressively.
> > > > > > > >
> > > > > > > > We refer to this **progressive expansion of the trust set** as Mechanism (i), and the **progressive reduction of the manipulated non-trust set** as Mechanism (ii). **The success of our DPTM relies on the simultaneous adoption of both mechanisms**, as:
> > > > > > > > * Mechanism (i) enables the SFDA model to learn real target domain features.
> > > > > > > > * Mechanism (ii) prevents features from the synthetic domain from becoming dominant, since there is an inherent gap that persists between the synthetic and real target domains (even though we employ alignments to bridge the gap).
> > > > > > > >
> > > > > > > > However, when we **forcibly assign all target samples to the non-trust set** (as in the DPTM-AN variant), the SFDA model is **denied access to any real target features during training**. Instead, it learns from synthetic samples exclusively. Due to the inherent domain gap between synthetic and real data (even when we use alignment strategies), the performance of DPTM-AN gradually stops increasing as $r$ increases. More fundamentally, this phenomenon stems from **the intrinsic and persistent domain gap between synthetic and real data**, a well-documented challenge that has been extensively recognized and empirically validated in prior work [8].
> > > > > > > >
> > > > > > > > [8] Akagic A et al. Exploring the Impact of Real and Synthetic Data in Image Classification: A Comprehensive Investigation Using CIFAKE Dataset. CoDIT 2024.

---

> > > > > > > > > ### Author Response · Authors · 2025-08-03
> > > > > > > > > **New Q4: The size of the target SFDA model (1/2)**
> > > > > > > > >
> > > > > > > > > ## New Q4: The size of the target SFDA model
> > > > > > > > >
> > > > > > > > > ## Reviewer comment
> > > > > > > > >
> > > > > > > > > In response to Q6, I don't understand why the authors gave me this answer. What does the Method in each row mean? Where’s Table A? Please review the first question again.
> > > > > > > > >
> > > > > > > > > ## Response
> > > > > > > > >
> > > > > > > > > We apologize for the confusion caused by our previous response and the lack of clarity regarding the table and model configurations. Below, we provide a detailed explanation to address the reviewer’s concerns.
> > > > > > > > >
> > > > > > > > > To better contextualize our understanding, we first quote the reviewer’s original comment:
> > > > > > > > > * When generating samples of the target domain and applying them to train the target SFDA model, it seems that the size of the target SFDA model will have a great impact on the performance. Is there any analysis related to this?
> > > > > > > > >
> > > > > > > > > In our experiments, **the target SFDA model** is **ResNet-50** for Office-31, Office-Home, and DomainNet-126, and **ResNet-101** for VisDA. Our understanding of this question is that the reviewer is asking whether **using a larger or smaller backbone architecture** would affect the performance of our method.
> > > > > > > > >
> > > > > > > > > To investigate this, considering we have already conducted experiments using ResNet-101 as the target model on the VisDA dataset, and the detailed results are presented in Table A.1 (where “Table A.1” refers to Table A.1 in the appendix in our original submission). To further analyze the impact of model size, we replaced the backbone with ResNet-50, which is smaller than ResNet-101, and conducted experiments on the same VisDA dataset (results are presented in the second row of the table in our initial response to Q6).
> > > > > > > > >
> > > > > > > > > Since the comparative methods did not report results using ResNet-50 on VisDA, we did not reproduce their results under this backbone, as doing so may result in underperforming versions of their performance and lead to an unfair comparison. Instead, to provide a meaningful reference for performance under ResNet-50, we include the result of directly evaluating the source model with ResNet-50 on VisDA (shown in the first row of the table in our initial response to Q6).

---

> > > > > > > > > > ### Author Response · Authors · 2025-08-03
> > > > > > > > > > **New Q4: The size of the target SFDA model (2/2)**
> > > > > > > > > >
> > > > > > > > > > For clarity, we reorganize all the key results as follows:
> > > > > > > > > > * The detailed **per-class performance** of our method and comparative methods using **ResNet-101** on the **VisDA** dataset is presented below. Each column (e.g., plane, bcycl) represents the classification accuracy for the corresponding class, while the last column reports the mean accuracy across all classes.
> > > > > > > > > >
> > > > > > > > > > | Method | Backbone   | plane | bcycl | bus  | car  | horse | knife | mcycl | person | plant | sktbrd | train | truck | Perclass |
> > > > > > > > > > |--------|------------|-------|-------|------|------|-------|-------|-------|--------|-------|--------|-------|-------|----------|
> > > > > > > > > > | Source | ResNet-101 | 92.3  | 33.3  | 76.4 | 60.9 | 86.5  | 32.7  | 89.9  | 33.3   | 79.8  | 48.0   | 87.7  | 18.4  | 61.6     |
> > > > > > > > > > | CPGA   | ResNet-101 | 95.6  | 89.0  | 75.4 | 64.9 | 91.7  | 97.5  | 89.7  | 83.8   | 93.9  | 93.4   | 87.7  | 69.0  | 86.0     |
> > > > > > > > > > | ASOGE  | ResNet-101 | 94.9  | 84.3  | 76.8 | 54.3 | 94.9  | 93.4  | 86.0  | 85.0   | 87.2  | 90.0   | 86.7  | 62.7  | 83.2     |
> > > > > > > > > > | ISFDA  | ResNet-101 | 97.5  | 91.4  | 87.9 | 79.4 | 97.2  | 97.2  | 92.2  | 83.0   | 96.4  | 94.2   | 91.1  | 53.0  | 88.4     |
> > > > > > > > > > | PS     | ResNet-101 | 95.3  | 86.2  | 82.3 | 61.6 | 93.3  | 95.7  | 86.7  | 80.4   | 91.6  | 90.9   | 86.0  | 59.5  | 84.1     |
> > > > > > > > > > | SHOT   | ResNet-101 | 95.0  | 87.4  | 80.9 | 57.6 | 93.9  | 94.1  | 79.4  | 80.4   | 90.9  | 89.8   | 85.8  | 57.5  | 82.7     |
> > > > > > > > > > | NRC    | ResNet-101 | 96.8  | 91.3  | 82.4 | 62.4 | 96.2  | 95.9  | 86.1  | 90.7   | 94.8  | 94.1   | 90.4  | 59.7  | 85.9     |
> > > > > > > > > > | GKD    | ResNet-101 | 95.3  | 87.6  | 81.7 | 58.1 | 93.9  | 94.0  | 80.0  | 80.0   | 91.2  | 91.0   | 86.9  | 56.1  | 83.0     |
> > > > > > > > > > | AaD    | ResNet-101 | 97.4  | 90.5  | 80.8 | 76.2 | 97.3  | 96.1  | 89.8  | 82.9   | 95.5  | 93.0   | 92.0  | 64.7  | 88.0     |
> > > > > > > > > > | AdaCon | ResNet-101 | 97.0  | 84.7  | 84.0 | 77.3 | 96.7  | 93.8  | 91.9  | 84.8   | 94.3  | 93.1   | 94.1  | 49.7  | 86.8     |
> > > > > > > > > > | CoWA   | ResNet-101 | 96.2  | 89.7  | 83.9 | 73.8 | 96.4  | 97.4  | 89.3  | 86.8   | 94.6  | 92.1   | 88.7  | 53.8  | 86.9     |
> > > > > > > > > > | SCLM   | ResNet-101 | 97.1  | 90.7  | 85.6 | 62.0 | 97.3  | 94.6  | 81.8  | 84.3   | 93.6  | 92.8   | 88.0  | 55.9  | 85.3     |
> > > > > > > > > > | ELR    | ResNet-101 | 97.1  | 89.7  | 82.7 | 62.0 | 96.2  | 97.0  | 87.6  | 81.2   | 93.7  | 94.1   | 90.2  | 58.6  | 85.8     |
> > > > > > > > > > | PLUE   | ResNet-101 | 94.4  | 91.7  | 89.0 | 70.5 | 96.6  | 94.9  | 92.2  | 88.8   | 92.9  | 95.3   | 91.4  | 61.6  | 88.3     |
> > > > > > > > > > | CPD    | ResNet-101 | 96.7  | 88.5  | 79.6 | 69.0 | 95.9  | 96.3  | 87.3  | 83.3   | 94.4  | 92.9   | 87.0  | 58.7  | 85.5     |
> > > > > > > > > > | TPDS   | ResNet-101 | 97.6  | 91.5  | 89.7 | 83.4 | 97.5  | 96.3  | 92.2  | 82.4   | 96.0  | 94.1   | 90.9  | 40.4  | 87.6     |
> > > > > > > > > > | DIFO   | ResNet-101 | 97.6  | 88.7  | 83.7 | 80.8 | 95.9  | 95.3  | 91.9  | 85.0   | 89.4  | 93.2   | 93.2  | 69.0  | 88.6     |
> > > > > > > > > > | ProDe  | ResNet-101 | 96.6  | 90.3  | 83.9 | 80.2 | 96.1  | 96.9  | 90.3  | 86.4   | 90.8  | 94.0   | 91.3  | 67.0  | 88.7     |
> > > > > > > > > > | **Ours (DPTM)**   | ResNet-101 | 99.5  | 97.1  | 96.2 | 93.0 | 99.2  | 99.2  | 98.8  | 97.7   | 99.3  | 99.7   | 98.6  | 93.1  | 97.6     |
> > > > > > > > > >
> > > > > > > > > > * The detailed per-class performance of the source model and our method using **ResNet-50** on the **VisDA** dataset is presented below. Each column (e.g., plane, bcycl) represents the classification accuracy for the corresponding class, while the last column reports the mean accuracy across all classes.
> > > > > > > > > >
> > > > > > > > > > | Method   | Backbone  | plane | bcycl | bus  | car  | horse | knife | mcycl | person | plant | sktbrd | train | truck | Perclass |
> > > > > > > > > > | -------- | --------- | ----- | ----- | ---- | ---- | ----- | ----- | ----- | ------ | ----- | ------ | ----- | ----- | -------- |
> > > > > > > > > > | Source   | ResNet-50 | 79.7  | 35.5  | 44.5 | 63.8 | 62.0  | 25.0  | 86.9  | 26.6   | 77.6  | 30.0   | 94.7  | 12.7  | 53.3     |
> > > > > > > > > > | **Ours (DPTM)** | ResNet-50 | 99.5  | 96.1  | 93.5 | 82.5 | 98.3  | 99.2  | 96.3  | 96.8   | 98.8  | 98.5   | 98.1  | 81    | 94.9     |
> > > > > > > > > >
> > > > > > > > > > These results demonstrate that:
> > > > > > > > > >
> > > > > > > > > > * Our method exhibits robustness to model size. It maintains high performance even when using a smaller target SFDA model ResNet-50. Notably, our method with ResNet-50 even outperforms existing comparative methods that use a larger ResNet-101 backbone, highlighting its superior adaptation performance regardless of model scale.
> > > > > > > > > >
> > > > > > > > > > * Our method is also scalable with respect to the target SFDA model size. When using a larger target SFDA model ResNet-101, our method achieves better performance compared to using ResNet-50, suggesting that its effectiveness can scale with increased target SFDA model size.

---

> > > > > > > > > > > ### Comment · Reviewer_pUPg · 2025-08-04
> > > > > > > > > > >
> > > > > > > > > > > Thank you for author’s responses and the detailed clarifications. I appreciate author’s efforts in addressing the concerns raised in my review. I have no further questions and will maintain my original score.
> > > > > > > > > > > I encourage authors to incorporate the relevant responses from the rebuttal into the final version of the manuscript to further improve its clarity and completeness.
> > > > > > > > > > > Thank you so much.

---

> > > > > > > > > > > > ### Author Response · Authors · 2025-08-07
> > > > > > > > > > > >
> > > > > > > > > > > > We thank the reviewer for the valuable insights and suggestions throughout the rebuttal process. We will incorporate the new experimental results and clarifications provided during the rebuttal into the revised version of the manuscript.

---

### Official Review · Reviewer_fShy · 2025-07-01

**Clarity:** 4
**Significance:** 3
**Originality:** 4
**Rating:** 5
**Confidence:** 3

**Summary:**

The paper presents DPTM, a generation-based method for source-free domain adaptation that leverages diffusion models to transform images from non-trust sets of the target domain into new images assigned to different categories while preserving target domain characteristics. The approach incorporates three core techniques: Target-guided Initialization, Semantic Feature Injection, and Domain-specific Feature Preservation. The method demonstrates strong performance across multiple benchmarks, outperforming state-of-the-art SFDA approaches in image classification tasks.

**Questions:**

1. **Can you provide comprehensive ablation studies showing DPTM's performance when each component (Domain-specific Feature Preservation, Target-guided Initialization, Semantic Feature Injection) is removed individually?**
2. **How does your targeted generation approach compare against methods that prioritize image diversity for target domain generalization, and can you provide experimental evidence supporting your design choice?**
3. **How does DPTM compare against other diffusion-based domain adaptation methods like DATUM for instance or other similar approaches, and what specific advantages do your multiple techniques provide over simpler generation approaches?**
4. **Can you provide more extensive qualitative results demonstrating the visual quality and domain adaptation effectiveness of your generated images?**

**Ethical Concerns:**

["NO or VERY MINOR ethics concerns only"]

**Final Justification:**

I appreciate the authors' thorough responses to my questions. They have addressed all concerns convincingly and went beyond the initial questions by offering additional insights into the effectiveness of their Progressive Refinement Mechanism. In light of these improvements, I am raising my score to accept.

**Limitations:**

yes.

**Paper Formatting Concerns:**

no concerns.

**Quality:**

4

**Strengths And Weaknesses:**

**Strengths:**

- **Writing Quality:** The paper is exceptionally well-written with clear, easily followable sections and well-motivated technical components.
- **Comprehensive Evaluation:** The experimental comparison against existing methods is extensive and thorough, providing strong empirical validation.
- **Technical Innovation:** The proposed techniques are ingenious in their dual approach of ensuring accurate generation of newly assigned classes while preserving essential domain-specific information within the images. The progressive refinement strategy represents a particularly compelling approach to the generation process.

**Weaknesses:**

- **Missing Component Analysis:** The method lacks quantitative ablation studies showing the individual impact of each technique. The paper should evaluate DPTM performance without Domain-specific Feature Preservation, Target-guided Initialization, and Semantic Feature Injection separately to understand each component's contribution.
- **Diversity vs. Target Alignment Trade-off:** While generating target domain-aligned images is beneficial, the paper doesn't address whether increased image diversity might provide better generalization to the target domain. The authors need to discuss this trade-off and demonstrate experimentally that naive targeted data generation performs worse than their proposed approach.
- **Incomplete Baseline Comparisons:** The paper omits comparison with existing generation methods that explicitly use diffusion models for pseudo-target domain population, such as DATUM [1]. Including these comparisons would strengthen the proposed technique and better justify the multiple components employed.
- **Limited Qualitative Analysis:** The authors provide insufficient qualitative results to fully demonstrate the effectiveness and visual quality of their generation techniques.

[1] Benigmim, Yasser, et al. "One-shot unsupervised domain adaptation with personalized diffusion models." CVPR 2023.

---

> ### Author Rebuttal · Authors · 2025-07-31
>
> ### Q1: Comprehensive ablation studies
> We have made ablation studies to evaluate each component on the Office-Home dataset, as reported below. The results demonstrate that all the components contribute significantly to the overall performance by obtaining average gains of 8.7%, 11.4%, and 9.0%, respectively.
>
> | Method                                   | Ar→Cl | Ar→Pr | Ar→Rw | Cl→Ar | Cl→Pr | Cl→Rw | Pr→Ar | Pr→Cl | Pr→Rw | Rw→Ar | Rw→Cl | Rw→Pr | Avg.  |
> |-|:-:|:-:|:-:|:-:|:-:|:-:|:-:|:-:|:-:|:-:|:-:|:-:|-|
> | Ours                                     |    86.7   |    94.2   |    92.8   |    91.5   |    94.0   |    92.6   |    90.6   |    86.4   |    92.8   |    90.5   |    87.1   |    94.7   |  91.2 |
> | w/o Target-guided Initialization         |    75.9   |    89.4   |    88.3   |    78.9   |    87.9   |    87.0   |    76.0   |    74.1   |    87.8   |    80.0   |    75.9   |    89.1   |  82.5 |
> | w/o Semantic Feature Injection           |    72.9   |    88.0   |    84.1   |    77.5   |    88.3   |    83.6   |    76.8   |    69.0   |    84.0   |    75.7   |    68.8   |    88.4   |  79.8 |
> | w/o Domain-specific Feature Preservation |    70.2   |    89.9   |    85.5   |    81.1   |    89.4   |    89.3   |    80.7   |    70.7   |    86.3   |    80.3   |    72.1   |    91.0   | 82.2 |
>
> ### Q2: Diversity vs. target alignment trade-off
> We focus more on target alignment. To explore the diversity vs. target alignment trade-off, we directly use text-to-image generation (which focuses more on image diversity) to replace our generation mechanism of non-trust set, and perform the same iteration refinement. Due to time limits for rebuttal, we perform experiments on the first 4 Office-Home tasks (Ar→Cl, Ar→Pr, Ar→Rw, and Cl→Ar). As shown below, our method prioritizing target alignment is superior to the method prioritizing image diversity.
>
> | Method        | Ar→Cl | Ar→Pr | Ar→Rw | Cl→Ar |
> |---------------|:---------:|:---------:|:---------:|:---------:|
> | Text-to-image |    66.6   |    87.5   |    86.4   |    82.1   |
> | Ours          |    86.7   |    94.2   |    92.8   |    91.5   |
>
> ### Q3: Incomplete baseline comparisons
> It is not feasible to directly compare our method with DATUM for two reasons. First, DATUM addresses UDA where source data is accessible, while we focus on SFDA without access to source data. Second, DATUM is for segmentation, while we specialize in classification.
>
> To address the concern, we include DM-SFDA [1] for comparison. As far as we know, DM-SFDA is the only existing generation-based SFDA method. It focuses on pseudo-source generation employing multiple stages of finetuning and generation. Since the source code of DM-SFDA is not available, we compare it with our methods on Office-31 (ResNet-50), Office-Home (ResNet-50), and VisDa (ResNet-101) by directly citing their results reported in [1]. The results demonstrate that our method significantly outperforms DM-SFDA on the majority of all SFDA tasks.
>
>
>
>
>
> | Method  | Office-31 |      |      |      |      |       |      |
> |---------|:---------:|:----:|:----:|:----:|:----:|:-----:|:----:|
> |         |    A→D    |  A→W |  D→A |  D→W |  W→A |  W→D  | Avg. |
> | DM-SFDA |    97.7   | 99.0 | 82.7 | 99.3 | 83.5 | 100.0 | 93.7 |
> | Ours    |    97.2   | 95.3 | 92.0 | 98.7 | 91.7 | 100.0 | 95.8 |
>
> | Method  | Office-Home |       |      |      |       |       |       |        |       |        |       |       |          |
> |---------|:-----:|:-----:|:----:|:----:|:-----:|:-----:|:-----:|:------:|:-----:|:------:|:-----:|:-----:|:--------:|
> |         |Ar→Cl | Ar→Pr | Ar→Rw | Cl→Ar | Cl→Pr | Cl→Rw | Pr→Ar | Pr→Cl | Pr→Rw | Rw→Ar | Rw→Cl | Rw→Pr | Avg.  |
> | DM-SFDA                                     |    68.5  |    89.6   |    83.3   |    70.0   |    85.8   |   87.4   |   71.3   |    69.6   |    88.2   |    77.8   |    68.5   |    88.7   |  79.5 |
> | Ours                                     |    86.7   |    94.2   |    92.8   |    91.5   |    94.0   |    92.6   |    90.6   |    86.4   |    92.8   |    90.5   |    87.1   |    94.7   |  91.2 |
>
>
> | Method  | VisDA |       |      |      |       |       |       |        |       |        |       |       |          |
> |---------|:-----:|:-----:|:----:|:----:|:-----:|:-----:|:-----:|:------:|:-----:|:------:|:-----:|:-----:|:--------:|
> |         | plane | bcycl |  bus |  car | horse | knife | mcycl | person | plant | sktbrd | train | truck | Perclass |
> | DM-SFDA |  98.1 |  89.8 | 90.6 | 90.5 |  96.8 |  95.2 |  92.2 |  93.4  |  97.8 |  94.4  |  92.4 |  48.8 |   86.3   |
> | Ours    |  99.5 |  97.1 | 96.2 | 93.0 |  99.2 |  99.2 |  98.8 |  97.7  |  99.3 |  99.7  |  98.6 |  93.1 |   97.6   |
>
>
> [1] Chopra S et al. Source-free domain adaptation with diffusion-guided source data generation. arXiv:2402.04929.
>
> ### Q4: Limited qualitative analysis.
> We conduct extra qualitative analysis in three parts: more generated samples, Grad-CAM of the SFDA model, and t-SNE for the SFDA model features. But according to the requirements of rebuttal, we will add them to the later version.

---

> > ### Comment · Reviewer_fShy · 2025-08-04
> >
> > I appreciate the authors' thorough response addressing my concerns. The provided explanations successfully resolved Q1, Q2, and Q4. However, regarding Q3, I believe the authors dismissed DATUM based on incorrect reasoning: while they state 'DATUM addresses UDA where source data is available,' this does not justify avoiding comparison with their generation pipeline, which operates independently of the source domain. Similarly, their claim that 'DATUM is for segmentation while we specialize in classification' overlooks the fact that DATUM's generation module is completely setting-agnostic, making the comparison still valuable.
> >
> > Given that DATUM's generation pipeline requires no source data and is domain-agnostic, and considering the authors have made their code publicly available, I maintain that a comparison with DATUM would be worthwhile

---

> ### Author Response · Authors · 2025-08-07
> **New Question: Comparison with DATUM (1/2)**
>
> ## New Question: Comparison with DATUM.
>
> ## Response
> We thank the reviewer for pointing out this issue. Upon further examination of DATUM, we agree that its generation module is also applicable to the SFDA setting and is task-agnostic. We apologize for the oversight in our initial response in not including a comparison with DATUM. We have now added a comparison between our method and DATUM as follows.
>
> We compare our method with DATUM on the Office-Home dataset. According to the DATUM paper, the method consists of three stages: (1) Employing training of DreamBooth to personalize the diffusion model by associating a unique token $V_∗$ with the appearance of the target domain. (2) Using the personalized diffusion model to generate a pseudo-target domain. (3) Training an existing UDA framework on the labeled source data and the generated unlabeled pseudo-target data. To align DATUM with the SFDA setting, we modify only the third stage. Specifically, we first train a source model using labeled source data, and then adapt the model to the target domain using the pseudo-target data generated by DATUM.  The results are shown below. For reference, we also include the performance of the source model as a baseline for comparison.
>
> | Method          | Ar→Cl | Ar→Pr | Ar→Rw | Cl→Ar | Cl→Pr | Cl→Rw | Pr→Ar | Pr→Cl | Pr→Rw | Rw→Ar | Rw→Cl | Rw→Pr | Avg. |
> | --------------- | ----- | ----- | ----- | ----- | ----- | ----- | ----- | ----- | ----- | ----- | ----- | ----- | ---- |
> | source          | 50.1  | 67.9  | 74.4  | 55.2  | 65.2  | 67.2  | 53.4  | 44.5  | 74.1  | 64.2  | 51.5  | 78.7  | 62.2 |
> | DATUM           | 55.3  | 76.8  | 79.3  | 65.1  | 77.7  | 78.6  | 62.4  | 52.1  | 79.7  | 66.6  | 55.9  | 80.5  | 69.2 |
> | **Ours (DPTM)** | 86.7  | 94.2  | 92.8  | 91.5  | 94.0  | 92.6  | 90.6  | 86.4  | 92.8  | 90.5  | 87.1  | 94.7  | 91.2 |

---

> ### Author Response · Authors · 2025-08-07
> **New Question: Comparison with DATUM (2/2)**
>
> The results demonstrate that:
>
> (1) **Compared with the source model**, DATUM achieves better adaptation performance, demonstrating its effectiveness in the SFDA setting. This result suggests that diffusion-based domain adaptation methods like DATUM are capable of significantly improving adaptation performance. We consider DATUM a valuable and insightful work, as its design showcases the potential of leveraging diffusion models to generate pseudo-target data for SFDA.
>
> (2) Compared with our method, the results demonstrate that **DATUM performs significantly worse than our method**. This may be due to the following two reasons:
>
> * A key factor contributing to the superior performance of our method is the use of our **Progressive Refinement Mechanism**, which enables the SFDA model to **progressively improve its performance through multiple iterations**. In contrast, DATUM lacks such a dynamic update mechanism, which may limit its adaptation ability.
>
> * We provide the detailed performance trajectory of our method as $r$ increases from 1 to 10. Notably, when $r = 1$, our method still outperforms DATUM. This demonstrates that **even without the Progressive Refinement Mechanism, our method remains more effective than DATUM**.
>
> | DPTM trajectory  | Ar→Cl | Ar→Pr | Ar→Rw | Cl→Ar | Cl→Pr | Cl→Rw | Pr→Ar | Pr→Cl | Pr→Rw | Rw→Ar | Rw→Cl | Rw→Pr | Avg. |
> |------------------|:---------:|:---------:|:---------:|:---------:|:---------:|:---------:|:---------:|:---------:|:---------:|:---------:|:---------:|:---------:|:----:|
> | $r=1$            |    60.2   |    80.9   |    82.8   |    67.8   |    79.3   |    80.4   |    64.5   |    57.1   |    81.6   |    69.1   |    60.5   |    83.4   | 72.3 |
> | $r=2$            |    72.6   |    87.2   |    85.7   |    73.5   |    85.4   |    84.4   |    72.3   |    69.7   |    85.2   |    76.7   |    70.8   |    87.7   | 79.3 |
> | $r=3$            |    75.3   |    89.7   |    87.6   |    77.7   |    88.9   |    86.4   |    79.2   |    76.1   |    86.7   |    80.1   |    75.5   |    90.2   | 82.8 |
> | $r=4$            |    79.0   |    91.8   |    89.1   |    81.0   |    91.2   |    88.3   |    82.1   |    79.1   |    88.1   |    82.7   |    79.1   |    91.7   | 85.3 |
> | $r=5$            |    81.5   |    92.5   |    89.8   |    83.9   |    92.0   |    89.4   |    85.3   |    81.5   |    89.3   |    84.5   |    81.8   |    92.8   | 87.0 |
> | $r=6$            |    83.7   |    92.9   |    90.7   |    86.1   |    92.8   |    90.2   |    87.1   |    83.1   |    90.0   |    87.4   |    83.3   |    93.3   | 88.4 |
> | $r=7$            |    85.0   |    93.7   |    91.5   |    87.5   |    93.0   |    91.2   |    87.8   |    84.4   |    91.0   |    88.6   |    84.9   |    93.9   | 89.4 |
> | $r=8$            |    85.9   |    94.2   |    91.7   |    88.7   |    93.8   |    91.9   |    89.0   |    85.1   |    91.7   |    89.7   |    85.6   |    94.2   | 90.1 |
> | $r=9$            |    85.9   |    94.1   |    92.0   |    89.8   |    93.9   |    92.3   |    89.8   |    85.6   |    92.2   |    90.2   |    86.6   |    94.4   | 90.6 |
> | $r=10$           |    86.7   |    94.2   |    92.8   |    91.5   |    94.0   |    92.6   |    90.6   |    86.4   |    92.8   |    90.5   |    87.1   |    94.7   | 91.2 |
>
> (3) The aforementioned results further indicate that the pseudo-target domain generated by our method is **better aligned with the real target domain** compared to that generated by DATUM.  We attribute this to the following reason. DATUM relies on DreamBooth to learn the appearance of the target domain and map it to a unique token $V_∗$. However, for classification tasks, **this process becomes challenging** when the true class labels of the target data are unknown. According to the paper of DATUM, during DreamBooth training, the lack of class labels forces the use of vague prompts such as "a photo of a $V_∗$ object". This may cause the learned token to not only capture the domain-specific appearance of the target data, but also absorb semantic information related to the object class and even irrelevant background features. As a result, the generated pseudo-target images may exhibit limited alignment with the true target distribution.

---

> > ### Comment · Reviewer_fShy · 2025-08-07
> >
> > I thank the authors for providing the comparison with the generation module presented DATUM. This comparison further strengthens the Progressive Refinement Mechanism. I therefore raise my score to 'Accept'

---

### Official Review · Reviewer_Rmwo · 2025-07-05

**Clarity:** 3
**Significance:** 3
**Originality:** 3
**Rating:** 5
**Confidence:** 4

**Summary:**

The paper proposes a method for Source-Free Domain Adaptation (SFDA) called Diffusion-Driven Progressive Target Manipulation (DPTM). The key idea is to leverage a pre-trained latent diffusion model (stable diffusion) to generate a pseudo-target domain from unlabeled target data. The target data are split into a “trust set” (high-confidence samples with reliable pseudo-labels) and a “non-trust set” (low-confidence samples) based on prediction entropy. The trust set samples are used directly with their pseudo-labels for supervised fine-tuning, while the non-trust samples are assigned new labels uniformly at random (to avoid class imbalance) and semantically manipulated into those categories using the diffusion model.

The manipulation through the diffusion model is carefully designed to preserve the target-domain characteristics of each image while changing its semantic class. DPTM employs a three-key strategy:
1.	Target-guided initialization: initialize the diffusion sampling with a latent that retains the original image’s structural features,
2.	Semantic feature injection: iteratively guide the diffusion process toward the assigned label,
3.	Domain-specific feature preservation: inject features from the original image into the diffusion latent to maintain the target domain style.

After this manipulation, the new generated images (with their assigned labels) together with the trust-set images form the “pseudo-target domain” used to train the target model. Importantly, DPTM is applied in progressive refinement iterations, i.e., after one round of adaptation, the target model is improved and can produce more confident pseudo-labels, so the trust set grows and the non-trust set shrinks.

Empirically, DPTM demonstrates state-of-the-art performances on 4 standard SFDA benchmarks of varying scales: Office-31, Office-Home, VisDA-2017, and DomainNet-126. It outperforms existing methods by substantial margins, especially on challenging transfers with large domain shifts.

**Questions:**

How exactly is the class condition provided to Stable Diffusion? The paper denotes conditioning on y, but since Stable Diffusion is typically text-conditioned, did the authors provide the class name as a text prompt? Or did they finetune an embedding for each class?

How is the training runtime of DPTM for each progressive refinement iteration or until producing good performances?

I’m still not fully convinced that DPTM benefits from the random assignment of the labels for non-trust samples as a starting point, instead of the pseudo-labeling mechanism like on the trust samples. Starting with completely random labels for the non-trust set might hinder reproducibility and stability. Could the authors clarify on this?

**Ethical Concerns:**

["NO or VERY MINOR ethics concerns only"]

**Final Justification:**

After considering all the responses in the rebuttal, my main concerns have been addressed, particularly
- Training complexity is on-par with a few other existing, advanced SFDA methods -- the trade-off between accuracy and complexity is still reasonable.
- Justification on applying random assignment of labels for non-trust samples through additional empirical observations.

Given that the performance gain induced by the proposed method is significant in the space of SFDA, I increase my final score to 5: Accept.

**Limitations:**

Yes, the authors mention that the proposed method is inherently constrained by the generation capability of the diffusion models.

However, some other potential practical burden of implementing such a multi-stage system is under-addressed.

It would be great to include a discussion on deployment challenges (e.g., memory/GPU needs, sensitivity to hyperparameters) and consider if the use of synthetic target features could inadvertently cause performance drift in critical systems.

**Paper Formatting Concerns:**

No major concern

**Quality:**

4

**Strengths And Weaknesses:**

**STRENGTHS**

**Originality**: The paper offers a creative take on SFDA by utilizing diffusion models for style-aware augmentation. The two-stage framework and spatial calibration mechanism are technically sound and well-motivated.

**Empirical performance**: In my opinion, this is the main strength of the paper. The proposed method produces strong results on 4 benchmark datasets with significant and consistent gains over prior work in the setting of SFDA. Extensive ablations demonstrate the contribution of each component.

**Clarity**: The high-level intuition behind each design component is clear, especially in describing the use of diffusion for feature stylization and progressive learning from clean to noisy pseudo-labels.

**WEAKNESSES**

**Practicality**: The method introduces multiple components (e.g., multiple loss terms, a style sampler trained using DDIM, a learned confidence regressor, a spatial calibration module), which may be cumbersome to train in practice and require careful tuning.

**Minor redactional issues**: Some parts of the paper suffer from broken English and formatting issues, e.g.,
- Abstract: “.. a challenging tasks …”
- Page 6 (experiments): “Comperative Methods”
- Appendix C: “Thess results demonstrate …”, “demnostrated”

---

> ### Author Rebuttal · Authors · 2025-07-31
>
> ### Q1: Practicality, sensitivity to hyperparameters, and performance in critical systems
> We appreciate the reviewer's insightful comment. In fact, we did not tune hyperparameters separately for each SFDA task. In lines 254-255, we intend to mean that we only explore different values of $E$ and $R$ once on Office-Home and fix $E$ to 0.01 and $R$ to 10 for all the experiments (except for ablation studies). The performance on benchmark datasets demonstrates that these hyperparameter values work for most cases. To validate the practicality of our methods in critical systems, we also tested the performance of our methods on real-world scenarios VLCS using unsupervised model selection without using target labels. We offer a detailed explanation in Response to Q1 by Reviewer mAks.
>
> ### Q2: Minor redactional issues
> We will correct these issues.
>
> ### Q3: Class condition
> Yes. We use the class name as the text prompt and use the pretrained Stable Diffusion without any finetuning.
>
> ### Q4: Time cost and memory needs
> We present a detailed analysis of the time cost and memory needs.
> 1. Time cost:
> The time cost can be mainly divided into two parts:
> * SFDA model training.
>   For any $r$-th refinement iteration with $r=1,2,...,R$, we train the SFDA model using the trust set and the manipulated non-trust set. The total number of samples of these two sets equals that of the target domain. We use supervised training here. The time cost mainly depends on the scale of the target domain, and remains constant as $r$ grows. We denote it as $t_{SFDA}$.
>
> * Sample generation.
>   For any $r$-th iteration refinement with $r=1,2,...,R$, the generation time for each sample remains constant, at about 1 second on a single NVIDIA Tesla V100. The generation time depends on the number of samples of the non-trust set, which is a subset of the target domain. Note that, with the growth of $r$, the size of the non-trust set declines rapidly, as shown in Figure 2. Thus, the total generation time for $R$ iterations equals the total number of generated samples $N_{total}$.
>
>   The total training time $t$ can be estimated by $t= R*t_{SFDA}+N_{total}$. We take the 12 tasks in the Office-Home dataset shown in Table 1 as examples. We set $E=0.01$ and $R=10$ and run the tasks on a single NVIDIA Tesla V100. It takes about **7.4 hours for SFDA model training** and **3.8 hours for generation** to **complete the total algorithm** on average for each task, demonstrating an acceptable time cost. Note that our model also allows parallel training on multiple GPUs to further reduce the time cost.
>
> 2. Memory needs:
> The memory needs can be mainly divided into two parts:
> * SFDA Model Training.
>   During any $r$-th refinement iteration, the memory needs remain constant and approximately equal to those of standard supervised learning on the target domain, and will not accumulate as $r$ grows.
> * Sample Generation.
>   Our approach generates samples sequentially (one sample at a time). Thus, this part only requires sufficient memory to run the Stable Diffusion model.
> In conclusion, our method does not require high total memory needs, and can be run comfortably on a single GPU with 24GB of memory.
>
> ### Q5: Random assignment of labels for non-trust samples.
>
> **From a performance perspective**, we employ random label assignment, instead of using the original pseudo-labels of non-trust samples given by the SFDA model, for two key reasons. First, there are no obvious patterns between the original pseudo-labels and ground truth labels of non-trust samples, and the original pseudo-labels cannot provide valid semantic priors. Second, if we use the diffusion model to semantically transform non-trust samples to their original pseudo-labels, this approach could inevitably introduce the class imbalance issue in the manipulated non-trust set. Therefore, we adopt a uniform random label reassignment strategy as formalized in Equation (2). To validate this claim, we perform comparative experiments using the original pseudo-labels for generation. Due to time limits, the experiments were performed on the first 4 Office-Home tasks (Ar→Cl, Ar→Pr, Ar→Rw, and Cl→Ar). The results presented below demonstrate the validity of our choice.
>
> | Method        | Ar→Cl | Ar→Pr | Ar→Rw | Cl→Ar |
> |---------------|:---------:|:---------:|:---------:|:---------:|
> |Original pseudo-labels    |    80.3   |    91.1   |    90.1   |    83.9   |
> | Ours (randomly assigned labels)          |    86.7   |    94.2   |    92.8   |    91.5   |
>
> From the perspective of reproducibility and stability, the results are fully reproducible as long as the random seed is fixed. In this case, the division between trust and non-trust sets becomes repeatable, and the sample ordering in the non-trust set's data loader is reproducible. Consequently, the new labels assigned to non-trust samples via Equation (2) are perfectly reproducible.

---

> > ### Comment · Reviewer_Rmwo · 2025-08-06
> >
> > I thank the authors for addressing my concerns and providing new insights / results in the rebuttal. I have also read the responses to other reviewers. I will increase my score accordingly.
> >
> > Please ensure that all the new insights and empirical results are incorporated into the revised manuscript or the appendix.

---

> ### Author Response · Authors · 2025-08-07
>
> We sincerely thank the reviewer for the valuable insights. We will incorporate the analysis and results into the revised version of the manuscript.

---

### Note · Authors · 2025-08-14

We sincerely thank all reviewers for their valuable insights. We are pleased to hear that our contributions were well-received. We also appreciate the comments highlighting the strengths of our work, specifically:
* **Strong and consistent empirical performance.** (Reviewer Rmwo, fShy, pUPg, LADn, and mAks)
* **Novelty, technical innovation, and clear motivation.** (Reviewer Rmwo, fShy, pUPg, and mAks)
* **Clarity and quality of presentation.** (Reviewer Rmwo, fShy, and LADn)

During the rebuttal process, **all reviewers agreed that we had satisfactorily addressed their concerns**. We sincerely thank them for raising these points, which help us further strengthen our work. Below, we summarize the shared concerns:

1. The **practicality and computational cost**.

As for practicality, we **addressed this concern** (Reviewer Rmwo, pUPg, and mAks) by:
* clarifying that our hyperparameters are fixed across all datasets, **avoiding per-task tuning**.
* validating our applicability in **real-world scenarios, including VLCS and TerraIncognita dataset** using **unsupervised model selection** without target labels, confirming robustness in practical usage.

As for computational cost, we **addressed this concern** (Reviewer Rmwo and pUPg) by:
* providing a runtime and memory analysis.
* demonstrating that the method runs efficiently on a single 24GB GPU and supports parallel acceleration.
* providing the **detailed runtime and memory cost** of our method and comparative methods on Office-Home and DomainNet-126.

2. The missing ablation studies of each component.

We **addressed this concern** (Reviewer fShy, LADn, and mAks) by detailed component-wise ablation studies on the Office-Home, clearly demonstrating the necessity and effectiveness of all components.

We will further refine our paper in the revised version to make our work clearer and more accessible to the broader research community. Specifically:
* Integrate **all new experiments and analyses** during the rebuttal process, with particular emphasis on addressing the aforementioned shared concerns.
* Add visual analyses, including Grad-CAM, t-SNE visualizations, and more generated samples.
* Correct all language and formatting issues.


We sincerely appreciate that several reviewers recognized our work’s potential to inspire the SFDA community. Our core idea can extend beyond SFDA to UDA and even scenarios without source models,  inspiring the broader DA community.

---

### Decision · Program_Chairs · 2025-09-17

**Decision:**

Accept (poster)

**Comment:**

The paper proposes a source-free domain adaptation (SFDA) method called diffusion-driven progressive target manipulation (DPTM) that builds a pseudo-target domain instead of synthesizing a pseudo-source. DPMT is validated empirically on the Office-31, Office-Home, VisDA, and DomainNet-126 image classification datasets and can outperform the SOTA. Five experts in the field reviewed this paper. Their post-rebuttal recommendations are 3 Borderline Accepts and 2 Accepts. Reviewers appreciated the technical novelty of DPTM, the clear presentation of manipulation components, and the convincing empirical results.

The authors provided a detailed rebuttal in response to reviews. The discussions and rebuttal clarified several concerns on class conditioning, computational complexity, fixed hyperparameters, ablations of individual modules, and comparisons to related diffusion-based SFDA approaches. These concerns were well addressed in the rebuttal, and three of the reviewers raised their ratings. Although remaining concerns remain, mostly to computational cost, and the need for more qualitative visualizations and comparisons with SOTA methods, the strengths outweigh the weaknesses. These were partly addressed in the rebuttal. Overall, this paper makes a decent contribution to the SFDA literature. I therefore recommend acceptance as a poster. The authors are encouraged to make the necessary changes to the best of their ability.